



Atmospheric Chemistry and Physics (acp)

# Formation of a bottomside secondary sodium layer associated with the passage of multiple mesospheric frontal systems

Viswanathan Lakshmi Narayanan[1], Satanori Nozawa[2], Shin-Ichiro Oyama[2,3,4], Ingrid Mann[1], Kazuo Shiokawa[2], Yuichi Otsuka[2], Norihito Saito[5], Satoshi Wada[5], Takuya D. Kawahara[6], and Toru Takahashi[7,8]

[1]Department of Physics and Technology, UiT – The Arctic University of Norway, Tromsø, Norway
[2]Institute for Space-Earth Environmental Research, Nagoya University, Nagoya, Japan
[3]Space Physics and Astronomy Research Unit, University of Oulu, Finland
[4]National Institute of Polar Research, Japan
[5]RIKEN Center for Advanced Photonics, RIKEN, Saitama, Japan
[6]Shinshu University, Japan
[7]Department of Physics, University of Oslo, Norway
[8]Electronic Navigation Research Institute, National Institute of Maritime, Port, and Aviation Technology, Tokyo, Japan

**Correspondence:** VL Narayanan (narayananvlwins@gmail.com)

**Abstract.** We present a detailed investigation of the formation of a secondary sodium layer at altitudes of 79-85 km below the main sodium layer based on sodium lidar and airglow imager measurements made at Ramfjordmoen near Tromsø, Norway on the night of 19 December 2014. The airglow imager observations of OH emission revealed four passing frontal systems that resembled mesospheric bores which typically occur in ducting regions of the upper mesosphere. For about 1.5 hours, the lower altitude sodium layer had densities similar to that of the main layer with a peak around 90 km. The lower altitude sodium layer weakened and disappeared soon after the fourth front had passed. The fourth front had weakened in intensity by the time it approached the region of lidar beams and disappeared soon afterwards. The column integrated sodium densities increased gradually during formation of the lower altitude sodium layer. Temperatures measured with the lidar indicate that there was a strong thermal duct structure between 87 and 93 km. Furthermore, the temperature was enhanced below 85 km. Horizontal wind magnitudes estimated from the lidar showed strong wind shears above 93 km. We conclude that the combination of an enhanced stability region due to the temperature profile and intense wind shears have provided ideal conditions for evolution of multiple mesospheric bores revealed as frontal systems in $OH$ images. The downward motion associated with the fronts appeared to have brought air rich in $H$ and $O$ from higher altitudes into the region below 85 km wherein the temperatures were also relatively high. This would have liberated sodium atoms from the reservoir species and suppressed the re-conversion of atomic sodium into reservoir species so that the lower altitude sodium layer could form and the column abundance could increase. The presented observations also reveal the importance of mesospheric frontal systems in bringing about significant variation of minor species over shorter temporal intervals.





## 1 Introduction

Metal layers exist in the upper mesosphere - lower thermosphere region of the Earth's atmosphere as a consequence of meteor ablation process (e.g. Plane et al., 2015, and references therein). One of the metal layers studied extensively is the sodium layer existing between 75 and 110 km altitudes. The altitudes of peak sodium density vary between 88 and 93 km depending on latitudes (Fussen et al., 2010). In high latitude winters, the peak altitudes are close to 88 km due to atmospheric circulation. On occasions, additional layers referred to as sporadic sodium layers (SSLs) are found in addition to the main sodium layer (Plane et al., 2015; Clemesha et al., 1996, 2004; Heinrich et al., 2008; Tsuda et al., 2015b; Takahashi et al., 2015; Miyagawa et al., 1999). The SSLs are defined as thin layers with peak densities at least twice that of peak density of main sodium layer. Generally, SSLs are observed above the peak of the main sodium layer.

Two important mechanisms that are proposed for formation of SSLs are briefly mentioned below. The sporadic E ($E_S$) layers are intense accumulation of plasma in the lower E region ionosphere. They form due to the wind shears collecting metal ion species in a narrow altitude range (Axford and Cunnold, 1966; Whitehead, 1970; Mathews, 1998). The neutralization of sodium ions from sporadic E layers is believed to be one of the foremost mechanisms in generation of SSLs. This appears to be particularly effective in high latitudes (Cox and Plane, 1998; Kirkwood and Nilsson, 2000; Qiu et al., 2016). The other important mechanism attributes the formation of SSLs to existence of higher temperatures. In elevated temperatures the chemical reactions occur faster liberating more sodium atoms from the reservoirs like sodium bicarbonate ($NaHCO_3$) and sodium hydroxide ($NaOH$) (Zhou et al., 1993; Zhou and Mathews, 1995; Plane, 2004). Often, the temperature enhancement is caused by breaking of gravity waves in the upper mesospheric heights (Ramesh and Sridharan, 2012). Apart from the SSLs, on occasions the existance of sodium and other metal species extends well into the thermosphere (Collins et al., 1996; Chu et al., 2011; Wang et al., 2012; Raizada et al., 2015; Tsuda et al., 2015a).

In this work, we present a case study investigating rare occurrence of a secondary sodium layer in lower altitudes below the main sodium layer. While secondary layers are sometimes noticed to occur in lower altitudes (Clemesha et al., 2004; Wang et al., 2012), they are not discussed in detail to our knowledge. The sodium layer formed below 85 km and its formation was consistent with passage of multiple mesospheric frontal systems. The frontal systems resembled mesospheric bores (described below) with an enhancement of $OH$ airglow intensities behind their passage. A total of four frontal systems were observed.

The mesospheric frontal systems are relatively infrequent waves exhibiting a boundary like feature in airglow images (Brown et al., 2004). They are often identified as mesospheric bores, which are nonlinear solitary type waves associated with a sharp discontinuity at the leading edge (Taylor et al., 1995; Dewan and Picard, 1998; Smith et al., 2003; She et al., 2004; Narayanan et al., 2009b, 2012; Dalin et al., 2013; Pautet et al., 2018). Usually there are phase locked undulations behind the leading bore front. Both theoretical and experimental studies indicate that the bores are similar to nonlinear waveforms generated due to the trapping and steepening of a long wavelength disturbance in a duct channel (Seyler, 2005; Yue et al., 2010; Narayanan et al., 2012; Grimshaw et al., 2015). There are also observations in which the leading bore front is followed by turbulence (known as





foaming bores or turbulent bores). This may happen when the amplitude of the bore is large such that it leads to wave breaking and formation of turbulence behind the leading bore front (She et al., 2004). In the mesospheric altitudes evidence for existence of bores in presence of both thermal and Doppler ducts are found (She et al., 2004; Narayanan et al., 2009b; Fechine et al.,

55   2009).

Airglow images often reveal either an enhancement or reduction of airglow intensities across the bore front. The bores are referred to as 'bright bores' ('dark bores') when the front is followed by enhanced (reduced) airglow intensity. The airglow enhancements (reductions) are attributed to the movement of the emission layer altitudes to higher (lower) temperature region (Dewan and Picard, 1998; Medeiros et al., 2005). Another interesting aspect of the bores when observed in multiple airglow

layers is the 'complementarity effect' between upper and lower airglow emissions. Complementarity effect is observation of intensity enhancements behind the front in the lower airglow layers with simultaneous intensity reductions in the upper airglow layers. This in phase and anti phase relationship between airglow layers appears to support the notion of bore jump occurring symmetrically with centroid situated between lower and upper airglow layers. The complementarity phase relations between different airglow layers depend on the location of the bore occurrence with respect to the airglow emission regions as discussed

in Dewan and Picard (1998) and Medeiros et al. (2005).

Sometimes, the mesospheric fronts are interpreted as leading edges of intense gravity waves that bring about noticeable changes in the upper mesosphere (Swenson et al., 1998; Batista et al., 2002; Li et al., 2007; Bageston et al., 2011). Such fronts are known as mesospheric wall waves. Such fronts do not necessarily show the nonlinear solitary type waves and further there can be considerable phase delays observed between the frontal passage in different airglow layers unlike mesospheric bores.

However, when the wave signatures are available in only one emission, it becomes difficult to identify whether the observed feature is a mesospheric bore or a wall wave. Hence the term 'mesospheric fronts' is preferred.

As mentioned before, we show the observations of formation of a lower altitude sodium layer concurrent with passage of multiple mesospheric fronts by comparing the sodium lidar and airglow imaging observations from the high latitudes. Next section provides information on datasets used. Results section explains the observations in detail which are discussed and

concluded in subsequent sections.

## 2   Data Used

### 2.1   Sodium lidar

We are going to discuss the optical observations made on the night of 19 December 2014 from Ramfjordmoen (69.6°N, 19.2°E) EISCAT radar site near Tromsø, Norway. A state of the art sodium temperature/wind lidar is operated during winter

periods from 2010. The lidar simultaneously transmits five beams and receives photons from mesospheric sodium by resonant scattering. The minimum temporal resolution of the data is 3 minutes with 96 m range resolution. In order to reduce noise level, data are averaged at 1 km vertical spacing. Further analysis is carried out based on data with 3 min temporal and 1 km altitude intervals. Further detailed information on the system is availabe in Nozawa et al. (2014) and Kawahara et al. (2017). On the night of observations the off-vertical beams were tilted at 12.5 degrees from zenith. It may be noted that the East - West





and the North - South beams are horizontally separated only by 35 to 45 km in the altitude range of 80 to 100 km, respectively. These beam separations are of the order of the horizontal wavelengths of high frequency gravity waves. Further, many of the high frequency waves propagate with velocities in excess of 50 m/s and their periods are typically less than few 10s of minutes (Pautet et al., 2005; Narayanan and Gurubaran, 2013; Suzuki et al., 2009, 2011). All these factors make the studies of the high frequency fast moving features from different beam measurements very difficult.

On the night of 19 December 2014, the lidar observations started first in the vertical beam by 13:35 UT (14:45 Central European Time) and finished by 08:00 UT on 20 December 2014. Intense clouds affected the measurements after 23:00 UT. The sodium densities are measured based on the resonant back scattering signal from individual beams along with an error estimate. We consider only those measurements with density errors less than 3% of the measured value. Except in the boundaries of the sodium layer the density error is less than 2% of the measured value. We have used the sodium density
data with temporal resolution of 3 min. The 96 m range resolved data are averaged to 1 km altitude steps. We mostly use the measurements from vertical beam and their column integrated values as will be discussed in later sections.

Temperatures are estimated individually from each of the beam. A corresponding error estimate is also made. We consider only those temperature values with errors less than 3% of the measured value. The temperature errors are generally less than 3 K except in the bottom- and top-most regions. We consider the average of all the temperatures from the five beams within
20 minute time duration and 1 km altitude interval as the background temperature in this work. This is done to smooth the fluctuations.

From the background temperature profiles we calculate the buoyancy frequency ($N$) at any altitude $z$ as given below.

$$N = \sqrt{\frac{g}{T}\left(\frac{dT}{dz} + \frac{g}{C_P}\right)} \tag{1}$$

where the acceleration due to gravity $g$ is taken as 9.54 m/s$^2$, $T$ represents the temperature and $C_P$ is the specific heat at constant
pressure taken as 1005 Jkg$^{-1}$K$^{-1}$. The temperature gradient in the above equation is calculated using center difference method. Buoyancy frequency is also known as Brunt-Väisälä frequency and is a complex quantity. The imaginary value of $N$ indicates that the atmosphere is convectively or statically unstable. A region of higher buoyancy frequencies bounded by the lower values both above and below is a potential thermal ducting zone. Gravity waves with vertical wavelengths nearly twice of the duct width are supposed to get trapped and intensified with constructive interference. This is typical gravity wave ducting due to
temperature profiles and known as thermal ducting (Hecht et al., 2001; Snively et al., 2010). However, in addition to the above wave ducting, intense thermal duct zones appear to produce nonlinear solitary type waves resulting in internal bores (Dewan and Picard, 2001; Grimshaw et al., 2015). The latter process leads to the formation of bores or fronts in the mesosphere. Hence, the existence of thermal ducting regions is important in the context of present work. Since $N$ is a complex value, we use $N^2$ to identify thermal ducts. Negative values of $N^2$ indicate regions of convective instability.

The most important reason for operating the lidar with five beams is to measure the horizontal winds along with vertical winds using the Doppler shift of the signal received. Line of sight winds and corresponding error estimates are made from the Doppler shift of the received signal. We consider only those measurements with error values less than 5 m/s for calculation of zonal, meridional and vertical winds. Values with larger error are neglected. The horizontal winds are estimated from the line





of sight winds measured by the off-vertical beams in the following way.

$$V_{mer} = (V_N - V_S)/2sin\theta_Z \quad ; \quad V_{zon} = (V_E - V_W)/2sin\theta_Z \tag{2}$$

In the equations, $V_{mer}$ and $V_{zon}$ respectively correspond to the meridional and zonal winds, $V_N, V_S, V_E, V_W$ represent the line of sight winds from the north, south, east and west beams, respectively. $\theta_Z$ stands for the zenith angle of the beams. Positive corresponds to eastward zonal and northward meridional winds while negative corresponds to westward zonal and southward meridional winds. Since the zonal and meridional winds are measured from off-vertical beams separated by about 35 to 45 km

in the altitude region of interest, it is assumed that the winds are spatially uniform over the region covered by the lidar beams. Similar to the temperatures, we consider 20 minute averages of zonal and meridional winds in 1 km vertical spacing as the background winds. From the estimated zonal and meridional winds in each altitude, we calculated the vertical shears of zonal and meridional winds respectively, applying central difference method. Positive (negative) wind shear values in the zonal or meridional directions indicate an increase of eastward or northward (westward or southward) wind magnitudes with height.

The vertical winds are obtained from measured Doppler shifts in the vertical beam. The positive values correspond to upward and negative to downward components. Since the vertical winds are smaller in magnitude and are often believed to be the result of waves, instabilities and turbulence, we do not make 20 minute averages for vertical winds. We consider the vertical winds in 3 minute temporal resolution with 1 km vertical resolution.

Another parameter we have estimated with the wind and temperature is the Richardson Number ($R_i$) given by the following

equation:

$$R_i = \frac{N^2}{(\frac{dV_{zon}}{dz})^2 + (\frac{dV_{mer}}{dz})^2} \tag{3}$$

here, $R_i$ is the ratio between static stability of the atmosphere given by $N^2$ and the wind shears given by the denominator of equation 3. It determines whether the atmosphere is stable or susceptible to the formation of instabilities. $R_i$ becomes negative only if the $N^2$ is negative and it indicates that the atmosphere is convectively unstable (or statically unstable). When $R_i$ is

negative it indicates definite presence of convective instabilities. Theoretical studies (Miles, 1961) showed that when $R_i$ is less than 0.25, dynamical instabilities (also referred to as Kelvin-Helmholtz or shear instabilities) can occur and turbulence can be maintained when $R_i$ is less than 1 (Hecht, 2004). This is due to the destabilization by the shears. However the upper limit for initiation of the instability is being debated (Hines, 1971; Majda and Shefter, 1998) and it is quite possible for the dynamical instabilities to form when $R_i$ is less than 0.5 (Li et al., 2005; Narayanan et al., 2012). Moreover, the existence of

low positive values of $R_i$ is only a necessary condition and in itself does not guarantee existence of dynamical instabilities. We have calculated $R_i$ to check whether the shears are large enough to create dynamical instabilities. Since the instabilities are usually short lived, we use data with 3 minute time resolution and 1 km vertical resolution to calculate $R_i$.

## 2.2 Airglow Imager

A collocated airglow imager is operated concurrently at the same location as a part of the Optical Mesosphere Thermosphere

Imagers (OMTI) network (Shiokawa et al., 1999). The imager has a deep cooled 512 x 512 pixel CCD sensor and is equipped





with a six position filter wheel with optical interference filters to study the following emissions: $OI$ 557.7 nm, $OI$ 630.0 nm, $OI$ 732.0 nm, $OH$ Meinel bands in the near infrared, $Na$ 589.3 nm and background sky intensity from 572.5 nm. All the emissions are measured with an exposure time of 30 s except for $OH$ and $OI$ 630.0 nm that are observed with 1 s and 45 s, respectively. Of interest to mesospheric studies are emissions from $OH$ Meinel bands, $Na$ and $OI$ 557.7 nm nightglow.

These airglow layers are known to emanate from a layer of approximately 10 km thickness with peak altitudes around ∼86 km, ∼90 km and ∼97 km respectively for $OH$, $Na$ and $OI$ 557.7 nm. On the night of 19 December 2014, $Na$ images were not acquired. The $OI$ 557.7 nm emission did not reveal any clear wave signatures. Therefore we are left with images from $OH$ emissions that are analyzed in detail. The $OH$ airglow images are available approximately once in every 2.5 or 3 minutes.

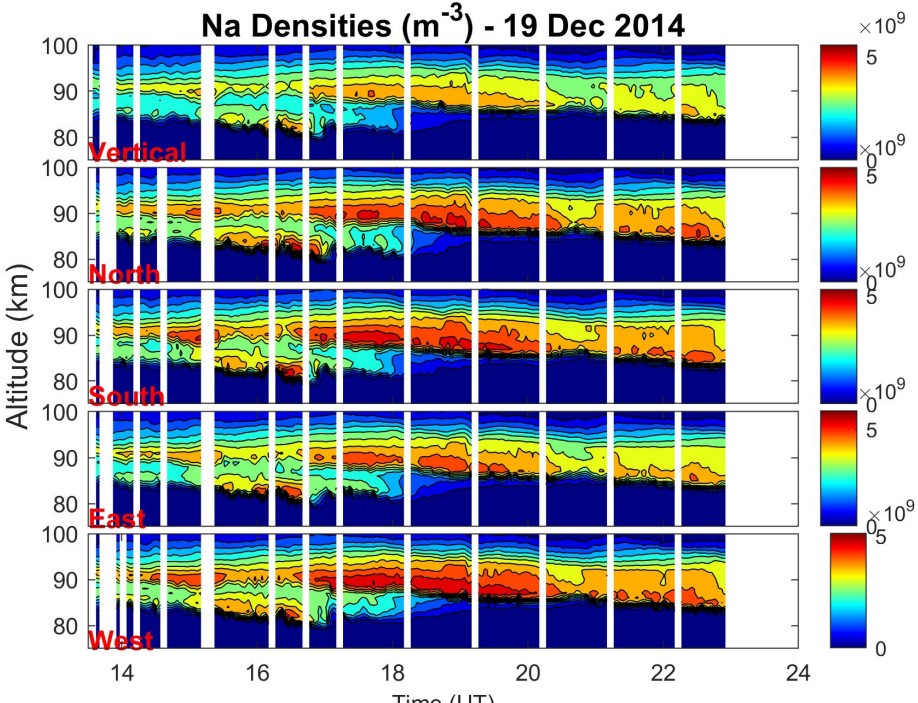

**Figure 1.** The sodium density measurements from five beam lidar showing formation of a secondary sodium layer below 85 km (Intense from 15:00 to 17:00 UT). The white regions indicate data gaps.

The $OH$ airglow images are 2 x 2 binned resulting in image size of 256 x 256 pixels. While binning increases the signal

to noise ratio in the images, the spatial resolution is compromised in the lower elevations. Therefore, we consider the region within 130° field of view for our analysis. The raw airglow images will be curved in the off-zenith regions because they are captured with a fisheye lens in the front end. There are well established unwarping procedures (Garcia et al., 1997; Narayanan et al., 2009a). We have unwarped the airglow images and projected them in equidistance grids so that the wave parameters can be properly estimated. For unwarping, we assumed that centroid altitude of $OH$ emission is at 86 km. Before unwarping, we

created percentage difference images ($I_P$) by subtracting and normalizing each individual image ($I$) with one hour running





average of images centered on the individual image ($\overline{\overline{I}}$) as given below.

$$I_P = \frac{I - \overline{\overline{I}}}{\overline{\overline{I}}} \times 100 \tag{4}$$

The wave events are visually identified and cross sections are extracted across the wave events. A strip of 25 pixel width is extracted across the wave signatures in the direction of their propagation. The strip is extracted from a region wherein the
waves are clearly seen and the average of the 25 pixels in each row of the strip results in the intensity cross section which is detrended afterwards. For a particular wave event, the same region is used for extracting the cross section from different images so that they can be used to estimate the phase velocities of the events. The distances along the cross section are obtained from the spatial coordinates of the pixels corresponding to the center of the strip. The distance value of the last point behind the location of wave events are subtracted so that the distances monotonically increase along the propagation direction of the wave.
The wavelength and phase velocity are obtained from the cross sections extracted from multiple images. The mean values of multiple estimations for wavelengths and phase velocities are provided with the standard deviations as errors. The propagation direction is measured within an accuracy of 2 degrees. North is assumed as 0 and East as 90 degree azimuths.

On the night of 19 December 2014, the airglow imaging observations were started at 14:40 UT and continued until 06:40 UT on 20 December 2020. However, useful mesospheric measurements were available only until 17:20 UT on 19 December
2020 as intense aurora occurred afterwards. The observations after 23:00 UT was hindered by the presence of clouds similar to the lidar measurements.

## 3  Results

### 3.1  Secondary sodium layer in lower altitudes

Figure 1 shows the sodium density measurements from the lidar. Note the occurrence of an intense lower altitude sodium
layer between 15:00 and 17:00 UT, which almost disappeared by about 18:00 UT. This pattern is similar in all the five beams and hence it is clear that the lower altitude sodium layer formed in an area larger than 35 square km, the minimum distance between the oppositely directed beams. To better investigate the time evolution of the lower altitude sodium layer, we show the 30 minute averages of sodium density profiles from the vertical beam in Figure 2. By the time of start of the lidar measurements at 13:35 UT, a secondary peak can be noticed already and it was located closer to the main sodium layer. Table 1 lists the
altitudes and peak densities of the main layer and lower altitude sodium layer. The peak of lower altitude sodium layer was initially at 87 km, separated by 4 km below the main sodium layer peak height. With time both the separation and intensity of the lower altitude sodium layer increased noticeably. After 15:30 UT, the peak density of lower altitude sodium layer was nearly as intense as the main layer at 91 km and its peak altitude has descended to 82 km, with a separation of 9 km to the peak of the main sodium layer. After 17:00 UT, the lower altitude sodium layer started to weaken and rapidly disappeared by
about 18:00 UT. Figure 2 clearly indicates that the lower altitude sodium layer was intense and well separated between 15:00 and 17:00 UT as already seen from the range-time-density maps of Figure 1.





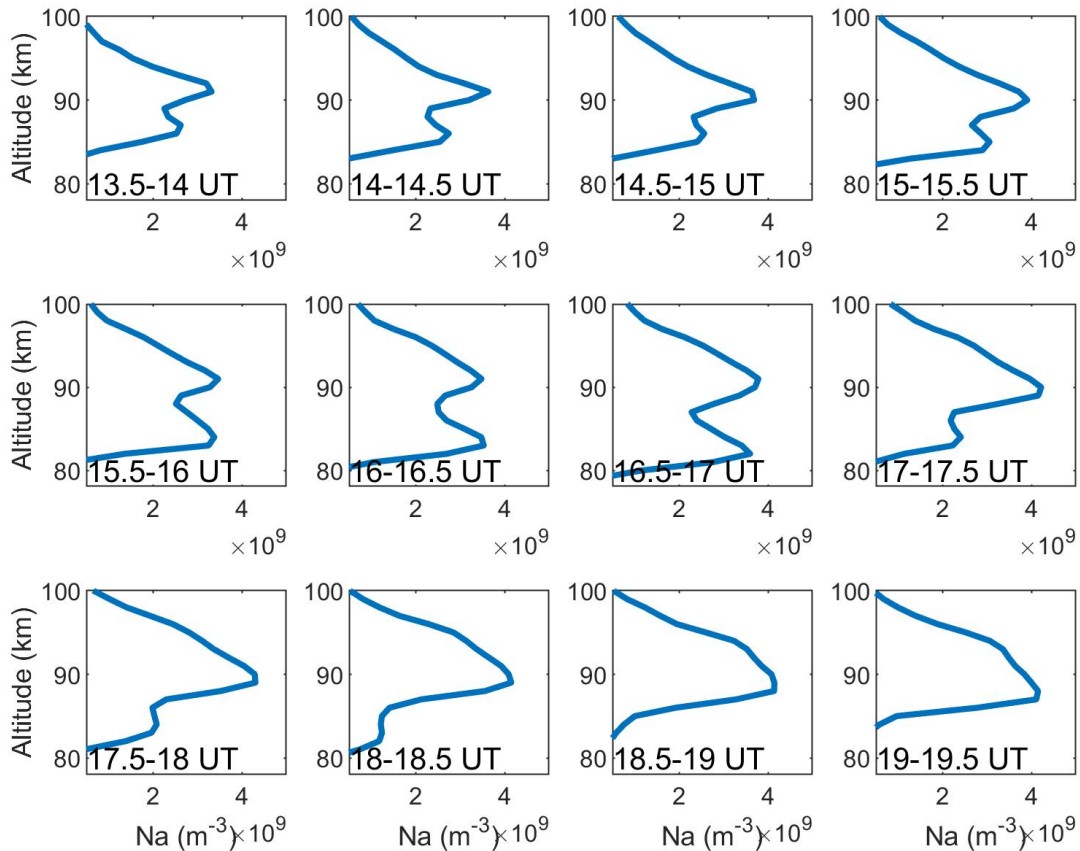

**Figure 2.** 30 minute averages of sodium densities from the vertical beam.

The variation of column abundance of sodium atoms is shown in Figure 3 along with the range-time-density plot from the vertical lidar beam. There was a gradual increase in the column abundance during the formation and intensification of the lower altitude sodium layer. This indicates that the lower altitude sodium layer is formed due to creation of sodium atoms from some reservoirs instead of from mere redistribution within the existing sodium layer. This is further consolidated by the observation that the column abundance was reduced after the disappearance of the lower altitude sodium layer. An increase in column integrated density of $1.85 \times 10^{13}$ atoms per $m^2$ occurred during the formation of the lower altitude sodium layer. There was approximately 50% increase in the total column abundance by 16:40 UT compared to the earlier hours around 14:00 UT.





**Table 1.** The peak altitudes and densities of main and lower altitude sodium layer

| Time (UT) | Main sodium layer | | Lower altitude sodium layer | |
| --- | --- | --- | --- | --- |
| | Altitude (km) | Peak density $\times 10^9$ (m$^{-3}$) | Altitude (km) | Peak density $\times 10^9$ (m$^{-3}$) |
| 13.5-14.0 | 91 | 3.3 | 87 | 2.6 |
| 14.0-14.5 | 91 | 3.6 | 86 | 2.8 |
| 14.5-15.0 | 90 | 3.7 | 86 | 2.6 |
| 15.0-15.5 | 90 | 3.9 | 85 | 3.1 |
| 15.5-16.0 | 91 | 3.5 | 84 | 3.4 |
| 16.0-16.5 | 91 | 3.5 | 83 | 3.5 |
| 16.5-17.0 | 91 | 3.8 | 82 | 3.6 |
| 17.0-17.5 | 90 | 4.2 | 84 | 2.4 |
| 17.5-18.0 | 89 | 4.3 | 84 | 2.1 |
| 18.0-18.5 | 89 | 4.2 | 83 | 1.2 |
| 18.5-19.0 | 89 | 4.1 | – | – |
| 18.5-19.0 | 88 | 4.2 | – | – |

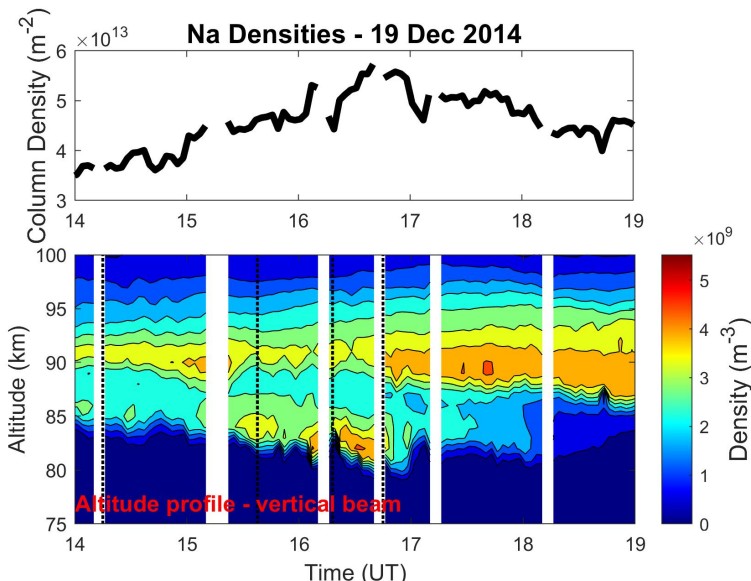

**Figure 3.** Sodium densities measured by the vertical beam. (Top) Column abundance of sodium atoms, (Bottom) range-time-density plot showing coincidence of column abundance variations with lower altitude sodium layer. The black dotted lines in the bottom panel indicate the estimated times of zenith passage of the fronts.





## 3.2 Passage of mesospheric frontal systems

The airglow imaging of $OH$ emissions indicates passage of multiple mesospheric frontal systems as discussed below. The airglow imaging started at 14:40 UT approximately one hour later than the start of lidar observations. The passage of an intense mesospheric front can be seen right from the first image acquired. A set of selected images on this night are shown in Figure 4. In the figure the fronts and wave observed on the night are marked with arrows indicating their direction of propagation. A corresponding movie is added as supplementary material showing the propagation of successive mesospheric

fronts approximately towards the top of the image frame corresponding to the North.

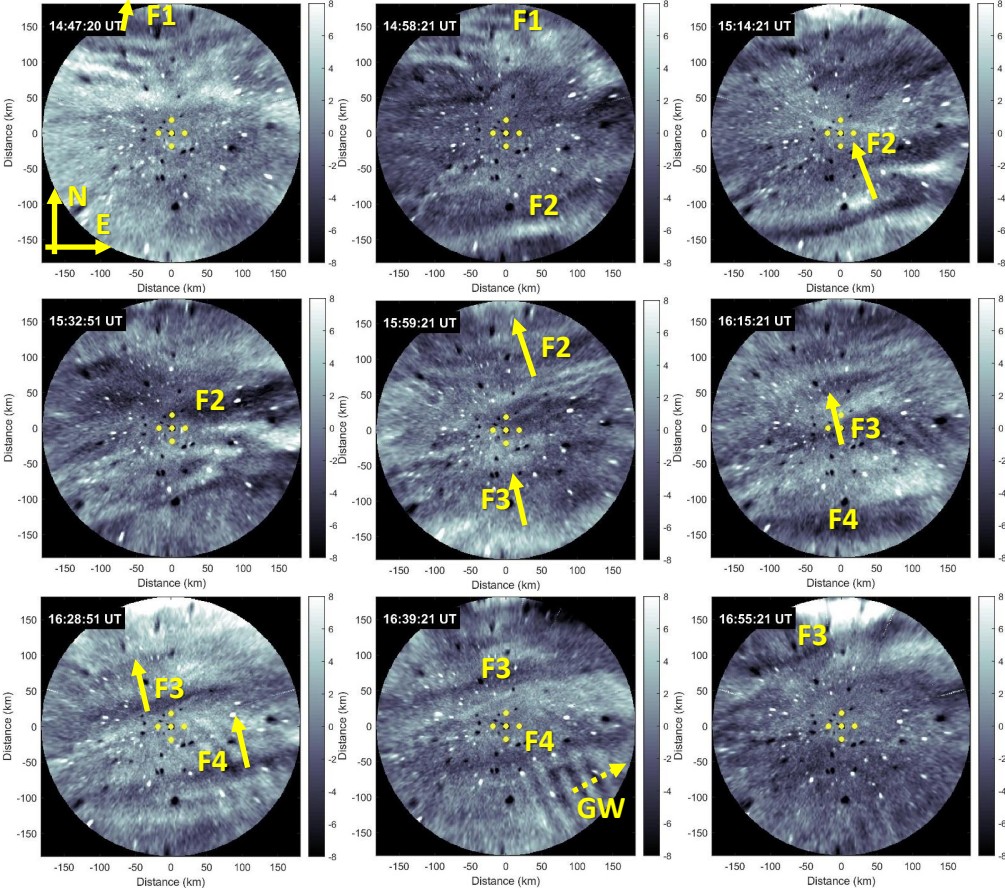

**Figure 4.** Selected $OH$ airglow images from 19 December 2014. Intensities represent the percentage perturbations according to equation 4. The yellow dots near zenith indicate location of lidar beams. The fronts are marked as F and their propagation direction is indicated by an arrow. The gravity wave is indicated by GW with dashed arrow.





As seen from Figure 4, the first front (F1) resembled an undular bright mesospheric bore with phase locked undulations behind the leading front. F1 was moving towards the North at an azimuth of ∼3 degrees. It has already crossed the zenith region of the observation site wherein the locations of lidar beams fall as indicated by yellow dots in the images. The estimated wavelength and propagation velocity of the feature are ∼23 km and ∼63 m/s respectively. Since the observations were started

at a later time, the probable time of zenith passage of F1 is estimated as 14:15 UT based on the calculated phase velocity. Figure 5(left) shows the cross sections extracted across F1 along its propagation direction. The F1 with trailing undulations can be seen clearly. Further, this front showed formation of additional wave fronts which is known to be a characteristic of the mesospheric bores (Dewan and Picard, 1998; Smith et al., 2003). As seen from Figure 5 the leading edge has reached the horizon around 15:00 UT.

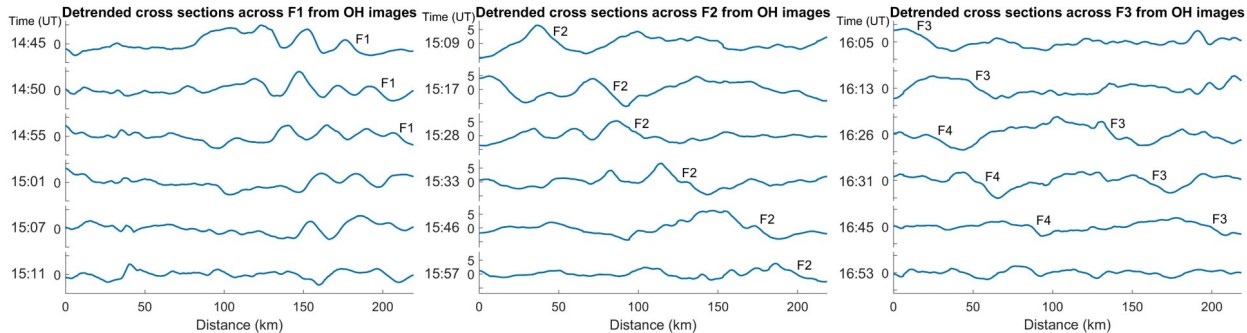

**Figure 5.** Cross sections extracted and detrended across the fronts from the percentage differenced $OH$ images. (Left) Cross sections across F1, (middle) across F2, (right) across F3 and F4.

The second front (F2) entered the imager field of view from the southern edge around 14:55 UT and it was also propagating predominantly towards the North with wave normal directed at ∼345°. There was an enhancement in $OH$ brightness behind F2. While F2 possessed trailing wave crests, the crests were not as defined as in F1. From the images at 15:14 UT and 15:32 UT in Figure 4, it appears like F2 shows some wave breaking signatures before 15:40 UT. Nevertheless, the cross sections enabled wavelength determination with a value of ∼31 km in the earlier part of its observation. F2 showed formation waves

with a smaller wavelength of ∼14 km after 15:45 UT. It continued to propagate in the same direction with an estimated phase velocity of ∼47 m/s. F2 has crossed the zenith region around 15:38 UT. Figure 5(middle) shows the cross sections across F2. The cross sections also reveal that the trailing undulations are initially with a larger wavelength and with smaller wavelength features after 15:40 UT (see last two rows in the middle panel in Figure5).

    The third front (F3) entered the images from the south around 15:56 UT and propagated across the zenith region by 16:23

UT. No clearly discernable phase locked undulations were found behind the front. The estimated phase velocity of this front is ∼87 m/s towards the North at an azimuth of ∼350°. F3 was the fastest of the observed fronts on this night. The fourth and final front (F4) observed on this night appears to follow the F3 along ∼350° azimuth. However, it was showing few weak phase locked undulations behind and was moving relatively slow. The estimated wavelength and phase velocity are ∼19 km and ∼63



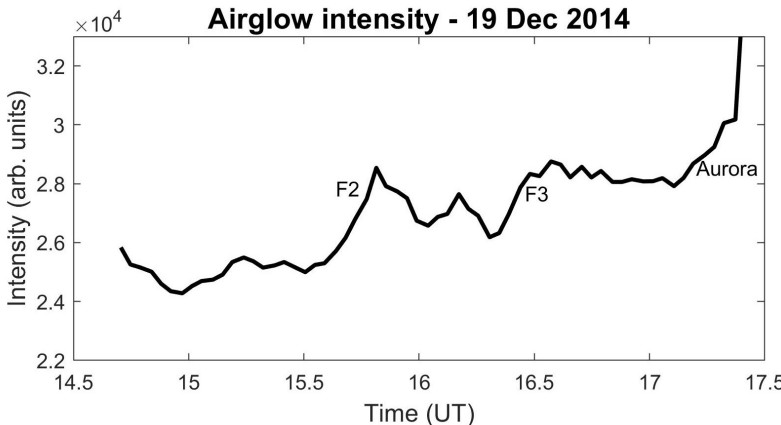

**Figure 6.** Zenith intensity time series from average of 16 x 16 pixels over zenith of the raw OH images.

m/s, respectively. Because of the differing phase velocities the distance between F3 and F4 has increased as can be verified

visually from images at 16:15 UT and 16:40 UT in Figure 4 and from cross sections in Figure 5. F4 became weak and almost unidentifiable in images after 16:45 UT, close to its passage over zenith. Table 2 lists the characteristics of the fronts. The apparent time periods of the fronts given in the table are obtained as the ratio between the wavelengths and phase velocities.

**Table 2.** The physical parameters of the observed fronts

| Fronts | Zenith crossing time (UT) | Direction (0°-North, 90°-East) | Wavelength (km) | Phase velocity (m/s) | Apparant time period (s) | Remarks |
|---|---|---|---|---|---|---|
| F1 | 14:15 | 3 | 23±4 | 63±11 | 365 | Imaging started by 14:40. Zenith crossing estimated based on phase velocity |
| F2 | 15:38 | 345 | 31±3 <br> 14±3 | 47±11 | 660 <br> 298 | The front was strong with some breaking signatures in the earlier times. After 15:45 UT, small scale waves evolved. Phase velocity appeared to decrease with time |
| F3 | 16:18 | 350 | – | 87±17 | – | No clear and consistent wave signatures behind the leading front to measure the wavelength |
| F4 | 16:44 | 350 | 19±3 | 63±16 | 301 | The front weakened close to zenith by 16:44 UT and not seen afterwards |





In addition to these four fronts, an east-northeastward propagating gravity wave disturbance is also noticed between 16:10 and 16:50 UT. In Figure 4, it is marked by GW with a dashed arrow indicating its propagation at 16:39 UT. Aurora started

to intensify from the Northern horizon in $OH$ images from around 16:55 UT. However, most parts of the images were clear to reveal wave signatures till about 17:20 UT. The auroral features extended southward and completely masked any wave signatures that could have occurred afterwards. This can clearly be seen from Figure 6 showing the zenith intensity time series. The intensities are averages of 16 x 16 pixels surrounding the zenith region of the raw $OH$ images. From Figure 6, it may be further verified that both F2 and F3 are accompanied by brightness enhancements in $OH$ images indicating that they are bright

bores. The sharp increase in intensity associated with aurora are seen from about 17:20 UT. It may be noted that there is no evidence in the past that the aurora enhances $OH$ Meinel band brightness directly. The observed enhancement is due to the entry of auroral light intensities through the broadband filter used to measure $OH$ Meinel band emissions. Nevertheless, it is worth noting that the signatures of the fronts and the gravity wave weakened significantly by 16:50 UT, well before the aurora masked $OH$ observations. This may also be seen from the last image in Figure 4 and the lowermost cross section in the right

panel of Figure 5.

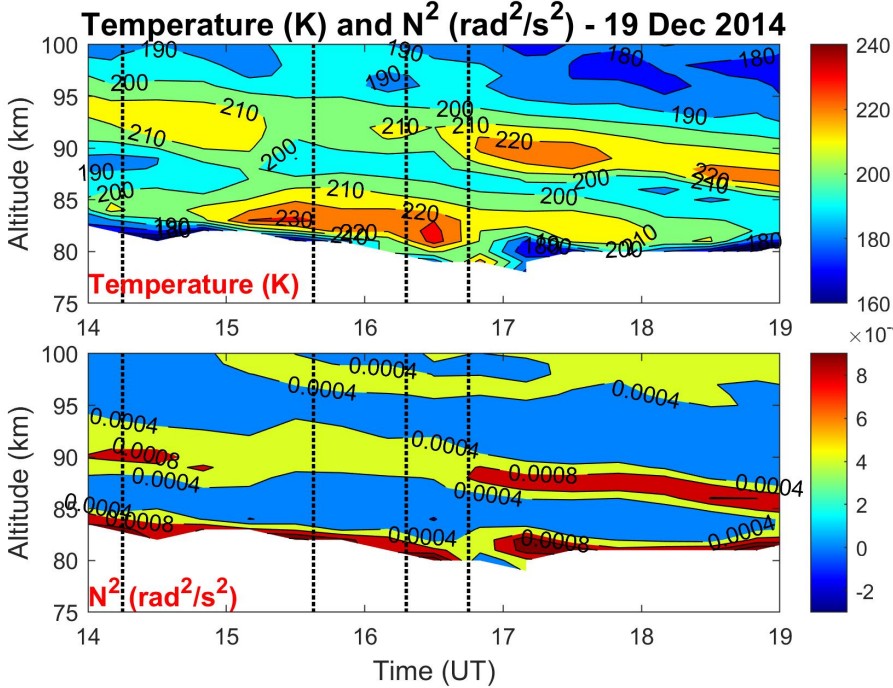

**Figure 7.** (Top) Background temperature structure, (Bottom) square of buoyancy frequency ($N^2$). The black dotted lines indicate the estimated times of zenith passage of the fronts.





### 3.3 Background temperature and wind conditions

Now we discuss the background temperature and wind conditions during the above mentioned observations. Figure 7(top) shows the background temperatures. The temperature profile shows a downward phase progression at the rate of 1 km/hr which might be due to the tidal effects. Of greater interest are the existence of an inversion layer close to 90 km altitude and enhanced temperatures in the lower altitudes coinciding with that of the lower altitude sodium layer. The temperature enhancement was particularly intense below 85 km in the duration between 15:00 and 17:00 UT. The square of buoyancy frequency profiles ($N^2$) estimated from the temperature profiles are shown in Figure 7(bottom). This profile clearly illustrates existence of an enhanced $N^2$ region bounded by lower values above and below. This region matches with the altitudes of thermal inversion. The enhanced $N^2$ region also shows a downward progression in concurrence with the downward progression in the temperature profiles, which is expected. Importantly, Figure 7 shows existence of a stable thermal ducting region on the night of 19 December 2014. Such regions can support the formation of mesospheric bores as mentioned in the introduction section. It may be seen that the width of the ducting region decreased after 17:00 UT. The location of the ducting region covers the peak region of main sodium layer prior to 17:00 UT.

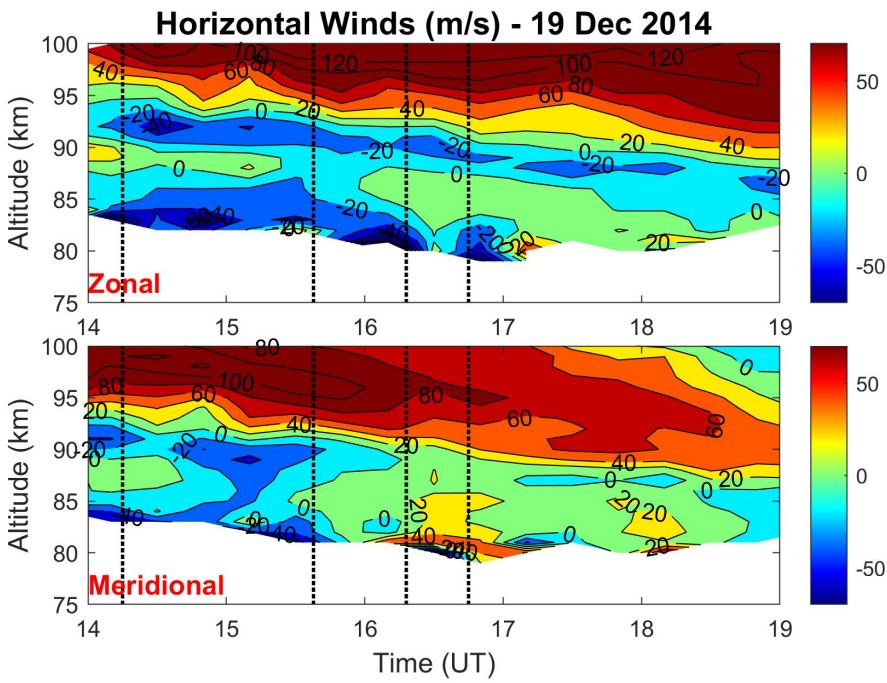

**Figure 8.** Measured winds. (Top) Zonal, (Bottom) Meridional. The black dotted lines show the estimated times of zenith passage of the fronts.





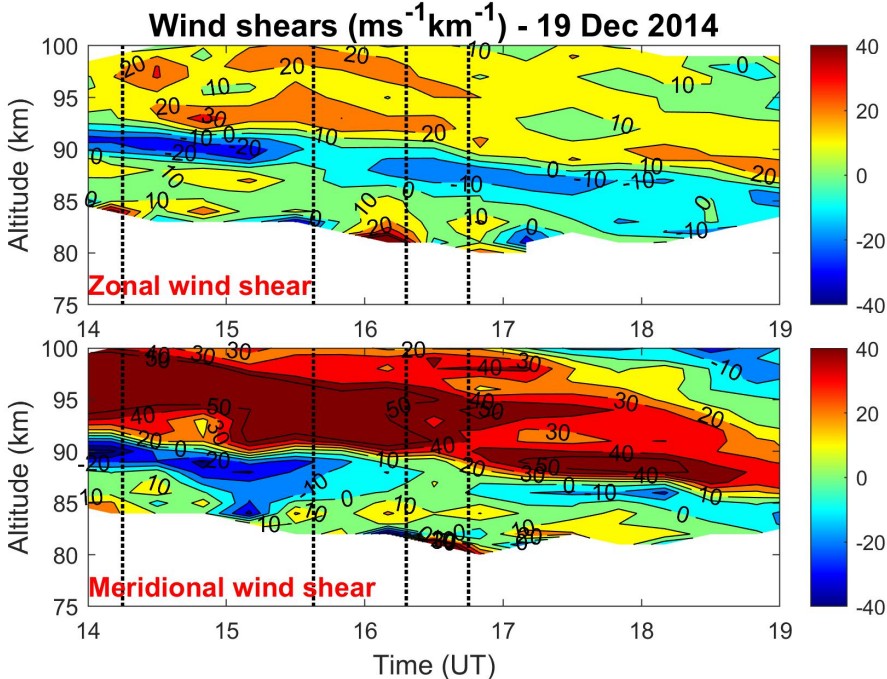

**Figure 9.** Vertical shears of horizontal wind. (Top) Zonal, (Bottom) Meridional. The black dotted lines indicate the estimated times of zenith passage of the fronts.

The zonal and meridional winds during the observation period are respectively shown in the top and bottom panels of

Figure 8. Since all the observed fronts propagated approximately to the North, the meridional winds approximately represent the background wind conditions along the propagation direction of the fronts. In the lower altitudes the wind velocities were within a magnitude of 40 m/s in both zonal and meridional directions. The meridional winds reversed from southward to northward around 16:00 UT in the altitudes between 83 and 93 km. There was a significant rise in the wind velocities with height, particularly above 93 km altitudes before 17:00 UT. The wind magnitudes increased by at least three times within a

narrow altitude region of about 4 km indicating existence of very high wind shears. The wind shears are shown in Figure 9. The shears in the meridional wind were much stronger than those in the zonal wind. Both Figures 8 and 9 show a downward phase progression very much similar to that seen in the temperature profiles. These large scale downward phase propagating features might be the result of tides.

Figure 10(top) shows the vertical wind measurements and Figure 10(bottom) shows the $R_i$ calculated from equation 3.

As mentioned in the previous section, the vertical winds and $R_i$ are shown in three minute temporal resolutions and only $R_i$ values below 0.25 are shown. Red dots indicate $R_i$ between 0 and 0.25 and blue dots indicate $R_i$ less than 0. Thus, red indicates probability for existence of dynamical instabilities and blue indicates definite occurrence of convective instabilities in the respective times and altitudes. Figure 10(top) also shows that the vertical winds were alternating frequently between



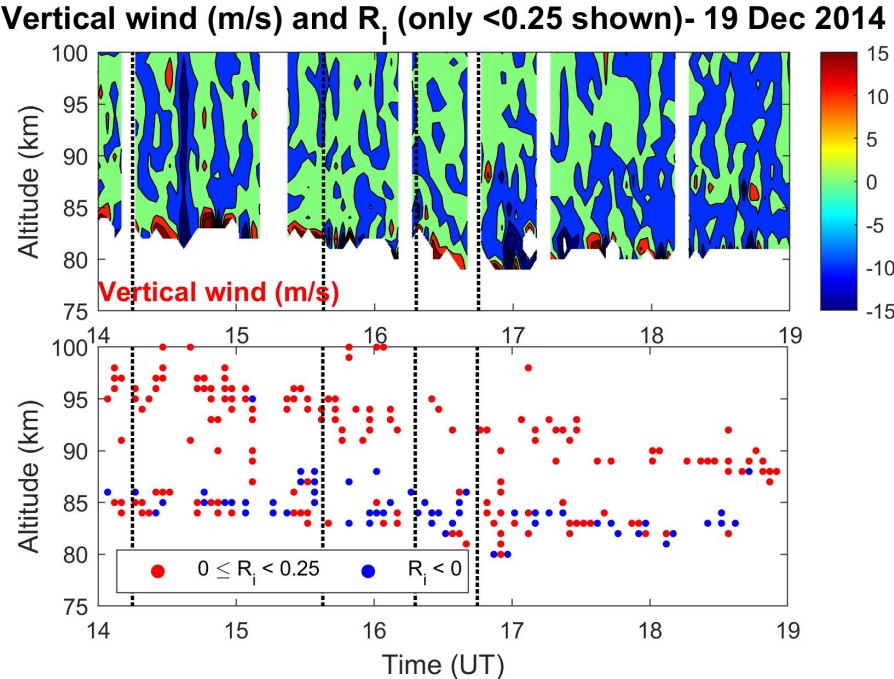

**Figure 10.** (Top) The vertical winds, (Bottom) Richardson number showing values between 0 and 0.25 as red dots and below 0 as blue dots. The black dotted lines in both the panels show the estimated times of zenith passage of the fronts.

positive and negative values, probably as a result of high frequency perturbations including the passage of fronts. We are
unable to isolate the contributions of the fronts from the vertical wind data. The vertical winds are expected to be upward directed in the regions of convective instability. A good match between the regions of convective instability in $R_i$ profile and higher values of vertical velocities can be noticed, for example around 14:45 UT near 85 km.

## 4    Discussion

The results described above lead to the following important observations that have to be explained. 1) There was a rare forma-
tion of sodium layer in altitudes below 85 km, 2) There was an enhancement in the column abundance of sodium atoms during the formation of this layer and the column abundance started to decline after its disappearance, 3) There were four consecutive mesospheric fronts coinciding with the duration of this lower altitude sodium layer observed with $OH$ images but not with $OI$557.7 nm images, 4) The mesospheric fronts were associated with enhanced $OH$ airglow intensities behind their passages and resembled bright mesospheric bores, 5) Temperatures were relatively higher in the region of lower altitude sodium layer,
6) There was a higher stability region indicating a thermal ducting structure matching with the altitudes of main sodium layer





located above the altitudes of lower altitude sodium layer, and 7) Horizontal winds had intense shears above 93 km along with a reversal in the meridional winds around 16:00 UT in the lower altitudes.

We have seen in Figures 4, 5 and 6 that the observed fronts are followed by regions of increased $OH$ airglow indicating that they might be mesospheric bores. F1 showed formation of new undulations as well. However, no clear signatures of the
fronts were seen in $OI$ 557.7 nm emission images. This is surprising given their intense signatures in the $OH$ emission region. The reason appears to be the existence of large wind shear in the region between $OH$ and $OI$ 557.7 nm emission layers as indicated by Figures 8 and 9. The fronts appeared to have disappeared owing to the critical level interaction between 93 and 95 km wherein the background wind speeds surpassed the speed of the observed fronts. This can be seen better with the help of Figure 11.

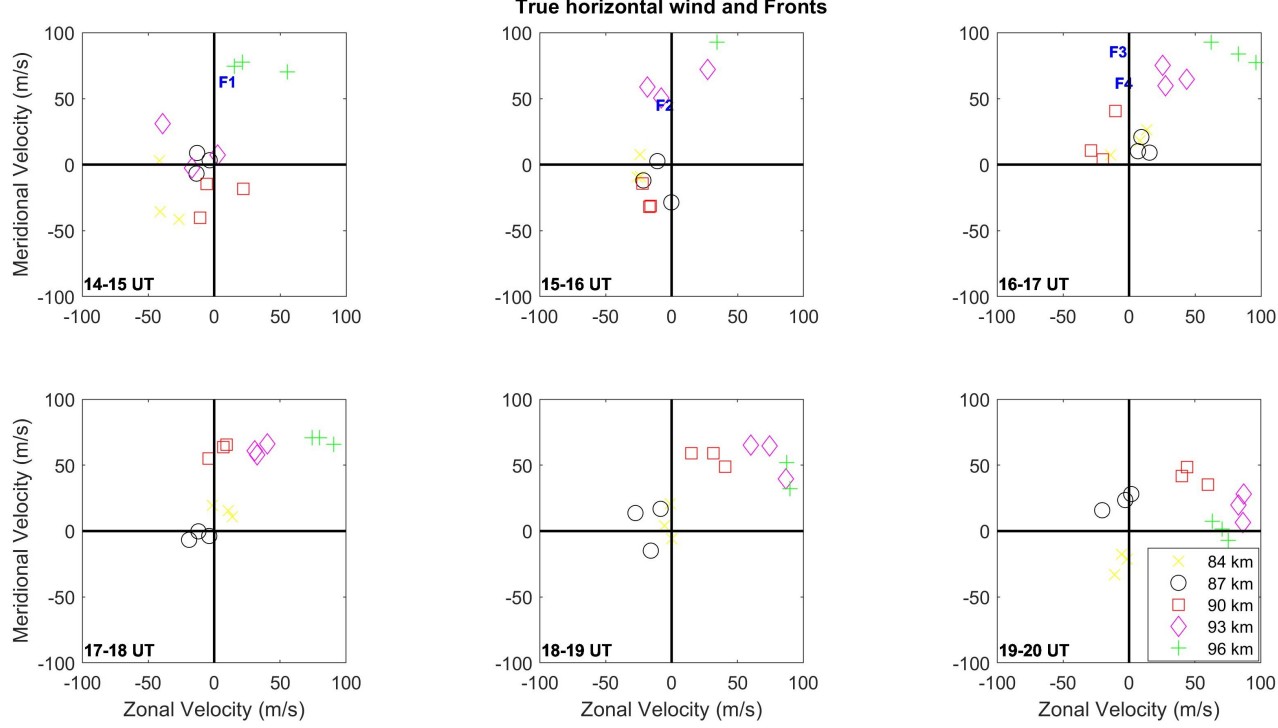

**Figure 11.** Horizontal wind measurements between 84 and 96 km in 3 km intervals along with the phase velocity of the observed fronts in the respective hours.

Figure 11 shows the horizontal winds measured by lidar in 3 km intervals at altitudes between 84 and 96 km in every hour. Since we use 20 minute averaged data, each height shows three points within every hour. This plot enables to identify the magnitude and direction of each of the 20 minute wind measurement within every hour in the respective altitudes. Also included are the observed velocities of the fronts in the corresponding hours when they were observed. The figure clearly illustrates that the background wind speeds were smaller than the speed of F1 below 93 km but were faster at 96 km. Therefore



critical level would have occurred in the region between 93 and 96 km restricting the front from perturbing the higher altitudes. Similarly for the case of F2, F3 and F4, the background wind speeds surpassed the fronts at $\sim$93 km altitudes resulting in their filtering from reaching higher heights. Particularly interesting is the case of F2. As mentioned earlier, F2 showed breaking signatures and after 15:45 UT revealed evolution of smaller wavelengths (see Figure 4 and corresponding discussion in section 3). The small scale features evolved on the crests of the front shortly after its zenith passage. It is likely that these features

are result of dynamical instabilities. Figure 10(bottom) shows the regions with $R_i$ less than 0.25 in altitudes of 90 to 95 km during this time. Therefore, dynamical instability would have occurred there resulting in the billow structures that could have perturbed the upper portion of $OH$ airglow layer. The existence of strong winds in the propagation direction of the fronts show the probable reason for not finding the waves in $OI$ 557.7 nm images.

The temperature profiles indicate that thermal duct was possible (Figure 7) and wind profiles indicate critical levels immedi-

ately above them (Figure 8). To check the possibility of ducting for the observed fronts, we calculate the vertical wavenumber profiles during the time of passage of the fronts over zenith with the following gravity wave dispersion relation (e.g. Narayanan and Gurubaran, 2013)

$$m^2 = \frac{N^2}{(u-c)^2} - \frac{u_{zz}}{(u-c)} + \frac{u_z}{H(u-c)} - \frac{1}{4H^2} - k^2 \tag{5}$$

In the above equation, $m$ and $k$ stands for the vertical and horizontal wave numbers respectively, $N$ is the buoyancy fre-

quency, $u$ and $c$ denote the background wind along the wave propagation direction and the phase velocity of the wave respectively, $u_{zz}$ and $u_z$ indicates $d^2u/dz^2$ and $du/dz$ respectively and $H$ is the scale height. Figure 12 shows the calculated $m^2$ profiles for each of the fronts with background conditions corresponding to the time of their passage over the zenith. Also lines at $m^2$ values indicating the vertical wavelengths of 3 km, 5 km and $\infty$ are shown. Since we do not know the horizontal wavenumber $k$ for F3, the last term in equation 5 is left out when calculating its vertical wavenumber $m$.

Generally, a wave undergoes reflection when the $m^2$ turns negative in a region. When there is a region of positive $m^2$ bounded by regions of negative $m^2$ above and below, the wave becomes ducted. A critical level occurs when the vertical wavelength of a wave approaches 0 and this will be seen as a sharp increase in the $m^2$ profile. Critical levels ensure that wave energy does not propagate beyond the level but they also contributes to stronger ducting at times. Strong wave reflection may happen when the critical level exist just above a region of stronger stability. In essence, existence of critical level at the top of

a duct results in a stronger duct with intense reflection of the wave because the leakage of energy through the duct is strongly restricted by the critical levels situated above (Lindzen and Barker, 1985; Skyllingstad, 1991; Ramamurthy et al., 1993). This has happened in the present case as can be inferred from Figure 12. In the altitudes below 86 km (90 km for F1), $m^2$ values become negative indicating the lower boundary of the ducting region. On the upper region, there is a very steep increase in the $m^2$ values indicating that strong winds cause critical levels. By comparing this figure with Figures 7 and 8, one can see

that the lower boundary is mainly due to the temperature profile and the upper boundary is caused by a combination of wind and temperature profiles (A similar case was observed for a lower atmospheric bore by Ramamurthy et al. (1993)). Hence, the important conditions required for formation of the bores were present on the night and the characteristics like enhanced airglow






**Figure 12.** The $m^2$ profile for the four fronts. Since F3 was not having trailing undulations, its $m^2$ is calculated leaving $k$ term in equation 5. The solid vertical line shows the 0 value and the dotted vertical lines show $m^2$ values corresponding to 5 and 3 km vertical wavelengths.

behind the fronts imply that these fronts were bores associated with a sudden downward push causing brightness enhancements in the underlying $OH$ airglow.

The increase of column abundance of sodium during the formation of lower altitude sodium layer is clearly seen from Figures 1 and 3. To investigate this further, we show in Figure 13 the total sodium column abundance along with the column integrated densities from 81 to 88 km corresponding to the region of lower altitude sodium layer, from 88 to 95 km corresponding to the main layer and from 95 to 102 km on the topmost region of atomic sodium layer in the upper atmosphere. All the densities shown are from the vertical lidar beam. As can be seen from the figure, the shape of the variations in total column abundance clearly matches with those of the integrated densities of the lower altitude sodium layer. The lower altitude sodium layer contributed to about 65% of the enhancement in the total column abundance. Remaining enhancement was due to the increase





in sodium concentration in higher altitudes. This is also revealed by the red and green lines in Figure 13. Interestingly, there was a reduction in the sodium densities around 15:30 UT in the altitude range of 88 to 95 km corresponding to the main layer. This time matches closely with the passage of F2 over the lidar beams.

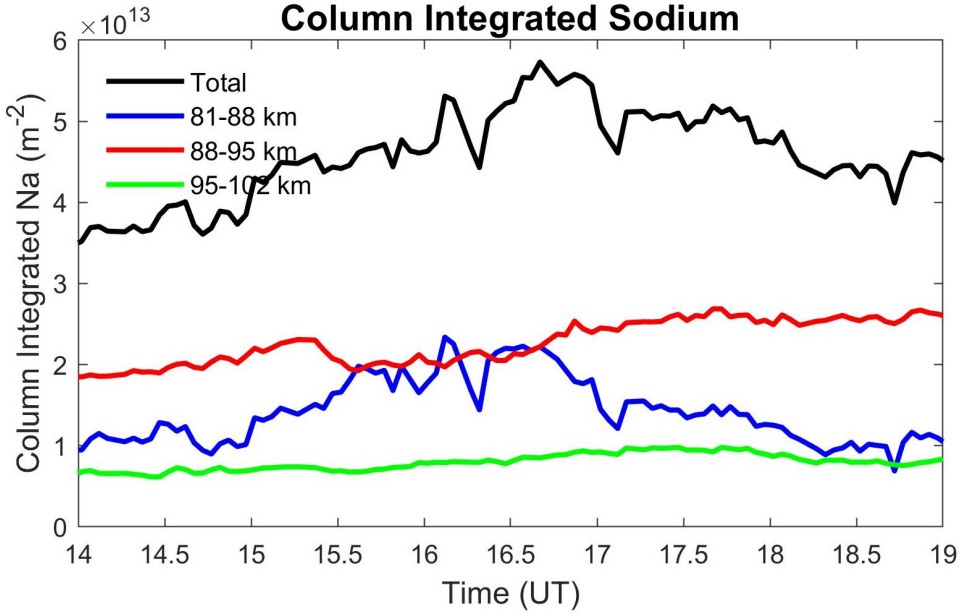

**Figure 13.** Column integrated sodium densities in selected altitude regions (blue and red lines) to compare with the total column abundance (black line)

The positive correspondence between the sodium density and temperature variations is already well known (Zhou et al., 1993; Zhou and Mathews, 1995). While there was relatively higher temperature in the region of lower altitude sodium layer below 85 km, it does not occur only on this day. Temperatures in the range of 220 to 250 K are fairly common below 85 km in winter months (Lübken and von Zahn, 1991; Nozawa et al., 2014; Takahashi et al., 2015; Hildebrand et al., 2017). To study the role of temperature in further detail, we show the sodium densities and averaged temperatures separately for

the height regions corresponding to lower altitude sodium layer (81-88 km) and the main sodium layer (88-95 km) in Figure 14. Note that we have used temperature data with 3 minute temporal resolution herein so that we can effectively compare them with the sodium density variations. Both densities and temperatures are 3 point smoothed in the plots. Figure 14(left) shows the integrated sodium densities and average temperatures for the region of lower altitude sodium layer from 81 to 88 km. The temperatures below 83 km are noisy resulting in large fluctuations. While there are some matching regions between

the densities and temperatures, the overall temperature variations differ from that of the sodium density in the lower altitude sodium layer region. For instance, the sodium densities continued to decrease while temperatures were nearly stable after 17:00 UT. This further indicates that the lower altitude sodium layer was not merely due to the temperature enhancement. However, the existence of higher temperatures in the lower altitudes is indisputable (see Figure 7(top)).





Figure 14(right) shows similar plots as above between the altitude region of 88 and 95 km corresponding to the main
sodium layer. Note that there was a temperature reduction just before 15:30 UT in this altitude region which is coincident with
the density reduction. F2 has crossed the zenith region around 15:35 UT. It is highly likely that this temperature reduction
corresponded to the signature of the passage of F2. The density reduction in the main sodium layer altitudes might therefore be
due to the sudden reduction in temperature associated with passage of F2. It is known that there may be phase delays between
the temperature and airglow intensity variations during passage of mesospheric bores (Taylor et al., 1995; Pautet et al., 2018).
For example, the very first report of a mesospheric bore by Taylor et al. (1995) had a temperature signature 15 minutes prior to
the passage of the bore.

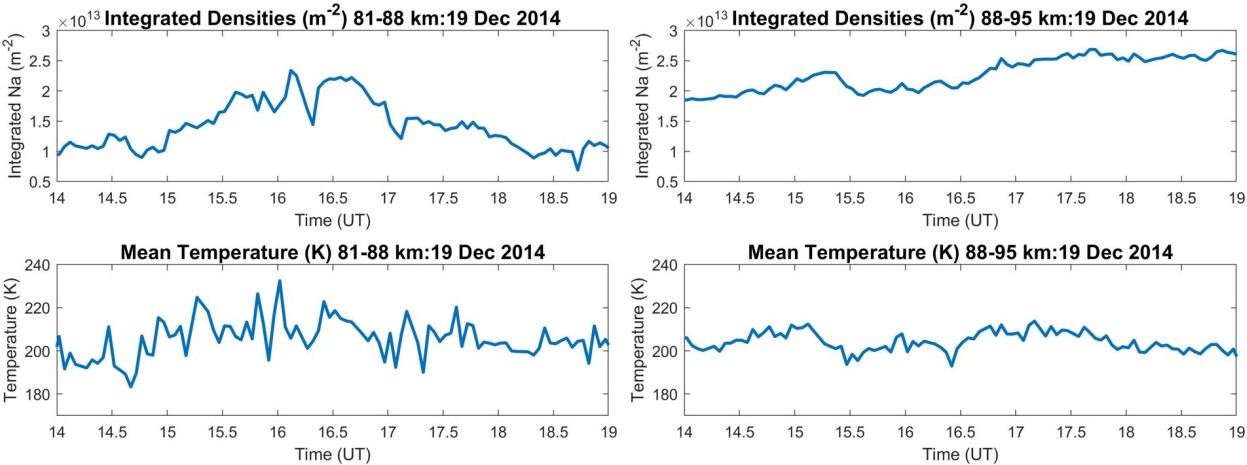

**Figure 14.** (Left) Integrated sodium densities and averaged temperatures from 81 to 88 km, (Right) Same as the previous one but from 88 to
95 km.

There was supposedly a downward force associated with the bright bores seen as fronts, which brings the minor constituents
from higher altitudes to lower altitudes. This is important for constituents like $O$ and $H$ whose mixing ratios increase with
altitude. Therefore, the downward transport will increase their concentrations in lower altitudes and affect the chemistry of
375 the region. Indeed such a downward force and associated movement is proposed as a reason for sudden intensity variations
following the bore jumps (Dewan and Picard, 1998). It is believed that these bores become bright in $OH$ emission because the
$OH$ emission peak moves to lower heights where temperatures are higher (Dewan and Picard, 1998; Medeiros et al., 2005). In
addition, such a downward movement also brings in $O$ and $H$ from upper altitudes to lower altitudes thereby increasing their
concentrations. This is because the mixing ratios of $O$ and $H$ increase with altitude in this region, as mentioned above. In this
380 case, the bores are supposed to have occurred in the region between 86 and 93 km (F1 appeared to have occurred a few km
higher). The start of the enhanced stability region associated with the temperature inversion around 86 km seems to determine
the lower boundary of the duct channel (see Figure 7). The upper boundary appears to be a combination of the temperature duct
along with intense wind shears causing critical level to the propagating wavelike structures. The bores would have occurred





near the center of the duct at ∼90 km from where the downward movement would have got initiated. In addition, it may be noted that the enhanced temperatures shown in Figure 7(top) at altitudes below 85 km also match well with the duration of observation of the fronts. There may be a contribution from adiabatic compression immediately below the altitudes of the duct due to the downward push casued by the bores. However, a detailed investigation on this aspect is beyond the scope of present work.

It is known that higher $H$ concentration occurring in the region with relatively higher temperature results in higher $OH$ emission rates as per the following reaction.

$$H + O_3 \rightarrow OH(\nu \leq 9) + O_2 \qquad\qquad 1.4 \times 10^{-10} exp[-470/T] \quad (cm^3 molecule^{-1}s^{-1}) \tag{R1}$$

The reaction R1 and its rate constant are taken from Smith and Marsh (2005). It can be seen that reaction R1 depends on temperature and higher temperatures result in higher reaction rates. Noteworthy is the fact that a downward push explains an enhancement in $OH$ airglow. On 19 December 2014, existence of the strong thermal ducting region coincident with the altitudes of main sodium layer would have favored formation of a mesospheric bore resulting in such a downward force and transport of minor species.

Now, we discuss how the sodium chemistry is affected by an increased concentration of $H$ and $O$ due to downward transport. At altitudes above 90 km, the densities and collisions are so low that formation of complex multi atomic molecules are often difficult. Further, during daytime higher EUV photon flux contributes to the dissociation of complex larger molecules. In lower altitudes, a larger portion of the sodium atoms react with other atoms and molecules and form reservoir species. The most important reservoir species for sodium is $NaHCO_3$, which liberates sodium atoms when interacting with $H$ (Plane, 2004; Plane et al., 2015).

$$NaHCO_3 + H \rightarrow Na + H_2CO_3 \qquad\qquad 1.84 \times 10^{-13}T^{0.777}exp[-1014/T] \quad (cm^3 molecule^{-1}s^{-1}) \tag{R2}$$

In addition $NaOH$ and $NaO$ can also liberate sodium as given below while interacting with $H$ and $O$, respectively.

$$NaOH + H \rightarrow Na + H_2O \qquad\qquad 4 \times 10^{-11}exp[-550/T] \quad (cm^3 molecule^{-1}s^{-1}) \tag{R3}$$

$$NaO + O \rightarrow Na + O_2 \qquad\qquad 2.2 \times 10^{-10}\sqrt{T/200} \quad (cm^3 molecule^{-1}s^{-1}) \tag{R4}$$

The reactions R2-R4 and corresponding rate constants are taken from Plane (2004); Gómez Martín et al. (2016). While the reactions R2-R4 are all dependent on temperature, the temperature dependence is weak for the reaction R4. The reaction R2 has a significant activation energy and hence is strongly dependent on temperature as can be seen from its rate expression.

Reactions R2 and R3 clearly show that more sodium can be liberated when atomic $H$ is transported from higher altitudes. There is sufficient atomic $H$ in the region between 80 and 90 km (e.g. Plane et al., 2015, Figure 4) so that a downward flux from 90 km region increases the concentration and mixing ratio of $H$ in lower heights. Though the mixing ratio of $NaHCO_3$ decreases with altitude in the region between 80 and 90 km, the lower altitude sodium layer forms in the region where the





concentration of $NaHCO_3$ is supposed to peak (Plane, 2004, Figure 5). Therefore, there will be sufficient concentrations of $NaHCO_3$ and $NaOH$ in the region and the rate with which the reactions occur will be higher when the temperature in the lower altitudes are higher. Since the temperatures were comparatively higher below 85 km more sodium atoms would have liberated resulting in a profound secondary sodium layer in the lower altitudes. Due to relatively lower values of temperature in the altitudes between 84 and 88 km, the amount of liberated sodium will be smaller in spite of the fact that the downward

$H$ flux was supposed to be present in those altitudes as well. In addition, in those heights the downward transport occurs in the region of decreasing mixing ratio of $NaHCO_3$, the most important reservoir species of sodium. This explains an apparent gap between the main sodium layer and lower altitude sodium layer.

The principle loss of sodium atoms below 85 km is through the formation of $NaHCO_3$, $NaOH$, $NaO$, $NaO_2$ and meteor smoke particles. However, atomic sodium undergoes only the following two reactions directly, whose products further react

with minor species in the mesosphere to produce more stable reservoirs like $NaHCO_3$.

$$Na + O_3 \rightarrow NaO + O_2 \qquad\qquad 1.1 \times 10^{-9} exp[-116/T] \quad (cm^3 molecule^{-1}s^{-1}) \tag{R5}$$

$$Na + O_2 + M \rightarrow NaO_2 + M \qquad\qquad 5.0 \times 10^{-30}[T/200]^{-1.22} \quad (cm^6 molecule^{-2}s^{-1}) \tag{R6}$$

The reactions and corresponding rates are taken from Plane et al. (2015). The reaction R6 decreases with increase in tempera-

ture and is of secondary importance compared to reaction R5. Therefore, in the region of lower altitude sodium layer wherein temperatures were higher, the removal of sodium atoms by $O_2$ was weaker.

The reaction R5 depends on $O_3$ concentration. The concentration of $O_3$ itself depends on the concentrations of $H$, $O$ and temperature. The production of $O_3$ is through the three body reaction given in reaction R7 which is inversely dependent on temperature. The generated $O_3$ is removed by $H$ through reaction R1 and $O$ through reaction R8. Both these reactions are

faster in higher temperatures. Reaction R1 is the major sink for $O_3$ during night times.

$$O + O_2 + M \rightarrow O_3 + M \qquad\qquad 6.0 \times 10^{-34}[300/T]^{2.4} \quad (cm^6 molecule^{-2}s^{-1}) \tag{R7}$$

$$O + O_3 \rightarrow O_2 + O_2 \qquad\qquad 8.0 \times 10^{-12} exp[-2060/T] \quad (cm^3 molecule^{-1}s^{-1}) \tag{R8}$$

The two reactions above and corresponding rates are from (Smith and Marsh, 2005). Therefore, the downflux of $H$ and $O$ to

the relatively higher temperature regions result in larger removal of $O_3$. This reduction of $O_3$ in turn affects the effectiveness of removal of generated sodium atoms through reaction R5. The above discussion indicates that sodium densities can increase when $H$ and $O$ are transported downwards when the temperatures in the lower altitudes are relatively high.

As mentioned earlier, we have seen from reaction R1 that the OH airglow intensity also increases when there is a downward flux of $H$ along with higher temperatures. All these observations indicate that the temperature and wind structure in the 86





to 93 km region lead to an intense ducting region wherein the observed mesospheric bores could have formed. The associated downward transport of $H$ and $O$ caused by the bores could have lead to the liberation of fresh sodium atoms in the lower altitudes from corresponding reservoirs according to reactions R2-R4. Further, the re-conversion of sodium to reservoir species would have been restricted due to the reduction in $O_3$ concentrations and relatively higher temperatures. This can explain the link between the observation of multiple mesospheric bores, formation of lower altitude sodium layer and enhancement in

sodium densities in the same duration. After the weakening and disappearance of the fronts, the downward transport of $H$ and $O$ would have stopped resulting in removal of sodium by regeneration of reservoir species from atomic sodium in the lower altitudes. Further, it may be noted that the temperatures below 85 km also decreased after the disappearance of the fronts.

Because we did not have airglow imaging observations before 14:40 UT and aurorae appeared after 17:15 UT, we are unable to probe the origins of the mesospheric fronts. Moreover, the focus of the present work is towards understanding the unusual

formation of lower altitude sodium layer and its relation to the observed mesospheric fronts rather than studying the formation and characteristics of the mesospheric fronts themselves. The lower altitude sodium layer occurred at altitudes that are too low for the ion chemistry to play any important role and hence we did not discuss ion chemistry associated with sodium production.

## 5    Conclusions

In this work, we discuss the sodium lidar and airglow imaging observations made on 19 December 2014 from Ramfjordmoen

(69.6°N, 19.2°E) near Tromsø, Norway. An unusual occurrence of a sodium layer below  85 km was noticed following the passage of four successive mesospheric frontal events observed in $OH$ airglow images (Figures 1-3). The fronts resembled bright mesospheric bores showing an enhancement in the $OH$ airglow intensity following their passage (Figures 4-6). The existence of a favorable ducting region for formation of the bores was present (Figure 12). Both temperature and wind profiles (Figures 7 and 8) contributed to the duct. The horizontal winds showed intense shearing region from ∼93 km altitudes (Figure

9). The critical levels occurring in this region restricted the propagation of the fronts to the $OI$ 557.7 nm airglow altitudes. The temperatures in the lower altitudes were in the range of 220 to 250 K during the formation of lower altitude sodium layer (Figure 7). While this magnitude of temperatures is not uncommon in the altitudes below 85 km, on this night the temperature enhancement coincided with the duration of the fronts. An enhancement in the column abundance of sodium was also seen to occur coincidentally with the formation of the lower altitude sodium layer (Figures 3 and 13). Further analysis showed that

the temperature alone cannot explain the formation of lower altitude sodium layer (Figure 14). We explain the observations consistently as follows.

The strong ducting appears to have provided favorable condition for formation of multiple mesospheric bores that are identified as the frontal features in $OH$ images. The downward transport of air rich in $H$ and $O$ associated with the mesospheric bores appeared to result in enhancement of $OH$ airglow intensity and release of atomic sodium from the reservoir species in

the lower altitudes. The existence of relatively higher temperature region below 85 km compared to the temperatures in higher altitudes could have led to increased reaction rates enabling larger release of sodium atoms from the reservoir species like $NaHCO_3$ and $NaOH$. Further the removal of atomic sodium by re-formation of reservoir species seemed to have reduced



under the conditions of enhanced temperature with downflux of $H$ and $O$. After 16:45 UT, the fronts weakened and disappeared thereby reducing the downward supply of the $H$ and $O$. Coincidentally, the temperatures below 85 km also lowered.
This could have resulted in re-conversion of the atomic sodium to sodium reservoir species which was seen as reduction in the column abundance of sodium and disappearance of the lower altitude sodium layer.

This event brings interesting new insights as below. 1) On rare occasions secondary sodium layers form below the main sodium layer. 2) The mesospheric bores play important role in altering the minor species concentrations in the mesospheric region within short temporal duration. 3) Multiple mesospheric bores can form with different phase velocities within few hours.

## 6 Data availability

The sodium lidar data used in this work is present in the website https://www.isee.nagoya-u.ac.jp/~nozawa/indexlidardata.html of ISEE, Nagoya University. Person responsible for the sodium lidar data is Dr. Satonori Nozawa. The all-sky airglow images used in this study are available at ISEE, Nagoya University from the website https://ergsc.isee.nagoya-u.ac.jp/data_info/ground.shtml.en. Person responsible for the airglow imager data is Dr. Kazuo Shiokawa. The 3 minute time resolution and 1 km altitude
averaged lidar data used for the calculations and illustrations in this work and the percentage differenced, equidistance projected OH airglow images are available in UiT website https://dataverse.no/dataset.xhtml?persistentId=doi:10.18710/C8MQ7V. A movie of processed OH airglow images is also provided in the webpage.

*Video supplement.* The time lapse of OH airglow images processed according to equation 4 in the manuscript is shown as a video in https://dataverse.no/dataset.xhtml?persistentId=doi:10.18710/C8MQ7V. The times of individual frames are shown as well. The colorbar
denote % perturbations.

*Author contributions.* VLN identified the event, carried out much of the analysis and prepared the manuscript. SN operated the sodium lidar and extracted the parameters from lidar measurement. SO supported operations of airglow imager and contributed to part of airglow image analysis. IM took part in the discussions and initiated the work along with discussions between VLN and SN. KS and YO are responsible for the airglow experiments and they took part in the discussion. NS, SW, TK and TT helped in setting up of the lidar experiment, supported the
maintanence and ensured its successful operation. All the authors took part in the discussion.

*Competing interests.* Authors have no competing interests

*Acknowledgements.* This work is supported by the Research Council of Norway through the grant NFR 275503. This study is partly supported by Grants-in-Aid for Scientific Research (17H02968) of Japan Society for the Promotion of Science (JSPS). A part of this work



is carried out while VLN visited Institute for Space-Earth Environmental Research (ISEE) under the International Joint Research program

of ISEE, Nagoya University. SO is supported by JSPS KAKENHI JP 16H06286, 15H05747, JPJSBP120194814 and Academy of Finland 314664. TT is supported by JSPS Overseas Research Fellowship. The sodium lidar has been operated at Ramfjordmoen, Tromsø, Norway under collaborations between ISEE, Nagoya University, RIKEN, Shinshu University, The University of Electro-Communications, EISCAT and Tromsø Geophysical Observatory, UiT. The airglow imager is operated at Ramfjordmoen, Tromsø, Norway by ISEE, Nagoya University. The instrumentation facilities are being supported by EISCAT faciliity at Ramfjordmoen, near Tromsø, Norway.



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
