# Peer review of "Formation of an additional density peak in the bottomside of sodium layer associated with the passage of multiple mesospheric frontal systems"

_Atmospheric Chemistry and Physics, 2020_

## Referee Comment (RC1) · Anonymous Referee #1 · 25 Nov 2020

This paper reports the observation of a secondary Na layer which formed around 85 km in the mesosphere during the passage a frontal system. The study involved a Na wind-temperature lidar which made measurements in the vertical and at 4 cardinal points, as well as an all-sky OH airglow imager. The imager was used to record the passage of four frontal events, and the lidar measured Na, wind and temperature. This data was combined to show that the front caused a marked temperature increase in a layer between 80 and 85 km, where the secondary Na layer then appeared. The wind and temperature data were also used to calculate the static and shear instability

surrounding the passage of each front.

The Na increase is interpreted to be caused by release of atomic Na from its reservoir NaHCO3, due to the higher temperatures which activate the reaction NaHCO3 + H, as well as downward transport of H and O from above 85 km and a corresponding decrease in O3. This interpretation seems quite plausible. Overall, this is a very nice piece of work which illustrates the importance of using multi-instrumented observations.

However, there are are several issues which the authors should address in a revised manuscript. The first is there must be a statement somewhere acknowleding the limitations of making observations in a Eulerian framework. That is, you are not observing the same air mass over 8 hours. This means that your interpretation of events requires that the atmosphere is horizontally homogeneous over roughly 2000 km. Whereas, in fact you only know the degree of homeneity over about 35 km (the distance between the off-zenith lidar beams), with some additional information over a larger scale from the all-sky imager. There is nothing you can do about this, but it should be stated in the paper.

The second issue is about the downward transport of H and O to below 85 km. From the way you describe this, the reader will imagine that the NaHCO3 reservoir is left unchanged below 85 km, to be joined by O and H from aloft. However, the NaHCO3 below 85 km will also be transported downwards. So it is actually a parcel of air containing NaHCO3, H and O from above 85 km that is transported downwards and heats adiabatically, releasing Na. Note that the mixing ratio of total Na increases with height up to the ablation peak of Na which is above 90 km (see recent papers e.g. Carrillo-Sanchez et al., (2020), Icarus, 335, art. no. 113395 ). So downward transport will also increase the total Na concentration (i.e. Na + reservoir species) below 85 km.

A third issue is that you list a large number of temperature-dependent rate coefficients, but do not do anything quantitative with them. That looks a little odd. For example, at

line 390 you provide the rate coefficient for H + O3, and state that this increases with temperature. But why not say how much? For example: "the rate coefficient increases by 40% when T increases from 200 to 230 K". That gives the reader some quantitative understanding of the point you are making.

None of these are major issues, and should be easily dealt with in a revised manuscript.

One other point - although the paper is well written and straightforward to read, there are many grammatical errors - particularly the absence of the definite article "the" and indefinite article "a". It is not the job of a reviewer to correct these basic errors.

Below is a list of minor corrections:

line 23: change to: "...as a consequence of meteoric ablation (e.g. Plane ....)"

line 24: the statement "In high latitude winters, the peak altitudes are close to 88 km due to atmospheric circulation." is not really correct - it is chemistry which determines the height of the Na layer; the role of circulation is principally in changing the local temperature profile.

line 41: "occur at lower altitudes"

line 73: "sodium lidar and airglow imaging observations from a high latitude location"

line 79: in what way is the lidar "state of the art"? Please specify. The performance parameters you mention sound fairly standard.

line 200: "This is further confirmed by the ..."

line 225: "...UT. The front continued ..."

line 269: "above 93 km before..."

line 314: "thermal ducting was possible"

line 319: "m and k stand for"

line 384: "would have been initiated"

line 399: in fact, the Na compounds (NaOH, NaHCO3 etc.) photolyse in the near-UV above 200 nm. So change EUV to UV.

line 446: "have led to "

The References need to be sorted out and checked. They are not all in alphabetical order, and the same author appears with different initials in difference references!

---

## Referee Comment (RC2) · Anonymous Referee #2 · 2 Dec 2020

This paper studies the formation mechanism of the secondary Na peak that appears within the altitude range of the main Na layer but below the main peak and near the bottomside of the main layer. The data quality (both lidar and airglow imager near Tromso) is high, and the analyses connecting the lidar-observed secondary Na peak below its main layer peak with the OH-imager-observed mesospheric bore event are extensive. The topic is interesting to the middle atmosphere science community. On this aspect, the paper is worth considering for publication in ACP after extensive reviews and revisions.

[Figure]

However, there are three major issues with the current manuscript: 1) The paper title is misleading or improper, 2) its Abstract reads badly with the first sentence distract people's attention, and 3) its Introduction contains misunderstanding of metal layer sciences.

All these issues likely stem from authors' misunderstanding of the meteoric metal layers. The main Na layer ranges from ~75 to 110 km, and the layer (below 85 km) they reported here is well within the main Na layer. Therefore, it is NOT an extra layer to the main layer, but an extra peak to the main layer peak. We have seen many times that Na layers go well below 85 km forming variable peaks during wintertime in the polar region, most likely caused by various wave activity. Therefore, what authors observed isn't new, but their studies of connecting such Na peaks to bore/frontal events are new and worth publishing.

1) Paper title: First, as written above it's not a new Na layer, but it's the secondary Na peak within the main Na layer; second, "in lower altitudes" has a grammar issue – lower than what? Therefore, such a paper title is not acceptable. Authors may consider to change the paper title to "Formation of an extra Na peak below the main layer peak associated with passage of multiple mesospheric frontal system" or something better.

2) Abstract: The first sentence in the Abstract is very misleading and it is frustrating to read it. Your paper is on the secondary Na peak below the main layer peak, but you started with mentioning something that is non-relevant to your subject. Please remove this sentence to avoid misleading readers. Also, change "additional sodium layer" to "additional Na peak".

3) Introduction: There is some lack of understanding of the thermosphere-ionosphere metal (TIMt) layers (mentioned in review paper by Plane et al. (2015)) in the Introduction, including thermosphere-ionosphere Fe and Na (TIFe and TINa) layers that were discovered to reach the altitudes of lower F region (Chu et al., GRL, 2011, 2020; Tsuda et al., GRL, 2015; Raizada et al., 2015; Chu and Yu, 2017). None of these pioneering papers were acknowledged. Instead, authors referenced Collins et al. (1996) and Wang et al. (2012), and adapted a bad phrase "double sodium layers". This "double sodium layers" phrase is improper and misleading, thereby it has been discarded by the field. Therefore, the current paragraph (the 3rd one in Introduction) is totally not acceptable. However, these TIMt layers aren't the focus of this manuscript, so authors may choose to remove this paragraph entirely and focus on the main Na layer. If authors want to include TIMt layers in the introduction, then they should update their understanding of the TIMt layers and cite proper references:

Chu, X., Nishimura, Y., Xu, Z., Yu, Z., Plane, J. M. C., Gardner, C. S., & Ogawa, Y. (2020). First simultaneous lidar observations of thermosphere‐ ionosphere Fe and Na (TIFe and TINa) layers at McMurdo (77.84°S, 166.67°E), Antarctica with concurrent measurements of aurora activity, enhanced ionization layers, and converging electric field. Geophysical Research Letters, 47, e2020GL090181. https://doi.org/10.1029/2020GL090181

Chu, X., Yu, Z., Gardner, C. S., Chen, C., & Fong, W. (2011). Lidar observations of neutral Fe layers and fast gravity waves in the thermosphere (110–155 km) at McMurdo (77.8°S, 166.7°E), Antarctica. Geophysical Research Letters, 38, L23807. https://doi.org/10.1029/2011GL050016

Raizada, S., Brum, C. M., Tepley, C. A., Lautenbach, J., Friedman, J. S., Mathews, J. D., et al. (2015). First simultaneous measurements of Na and K thermospheric layers along with TILs from Arecibo. Geophysical Research Letters, 42, 10,106–10,112. https://doi.org/10.1002/2015GL066714

Tsuda, T. T., Chu, X., Nakamura, T., Ejiri, M. K., Kawahara, T. D., Yukimatu, A. S., & Hosokawa, K. (2015). A thermospheric Na layer event observed up to 140 km over Syowa Station (69.0°S, 39.6°E) in Antarctica. Geophysical Research Letters, 42, 3647–3653. https://doi.org/10.1002/2015GL064101

Chu, X., & Yu, Z. (2017). Formation mechanisms of neutral Fe layers in the

thermosphere at Antarctica studied with a thermosphere-ionosphere Fe/Fe+ (TIFe) model. Journal of Geophysical Research: Space Physics, 122, 6812-6848. https://doi.org/10.1002/2016JA023773

Considering all these factors above, I rate the paper's scientific significance as "excellent", scientific quality as "fair", and presentation quality as "fair".

---

## Author Comment (AC1) · 2 Dec 2020

Response to the reviewer comments on "Formation of a bottomside secondary sodium layer associated with the passage of multiple mesospheric frontal systems"

We thank the reviewer for the time spent on the review and very useful suggestions for further improvement of the manuscript. Below, we give our responses to the reviewer's comments. Reviewer's comments are given between double backslashes and our responses follow below the comments.

[Figure]

(i.e \\ Reviewer Comment \\

Our replies ).

\\ This paper reports the observation of a secondary Na layer which formed around 85 km in the mesosphere during the passage a frontal system. The study involved a Na wind-temperature lidar which made measurements in the vertical and at 4 cardinal points, as well as an all-sky OH airglow imager. The imager was used to record the passage of four frontal events, and the lidar measured Na, wind and temperature. This data was combined to show that the front caused a marked temperature increase in a layer between 80 and 85 km, where the secondary Na layer then appeared. The wind and temperature data were also used to calculate the static and shear instability surrounding the passage of each front. The Na increase is interpreted to be caused by release of atomic Na from its reservoir NaHCO3, due to the higher temperatures which activate the reaction NaHCO3 + H, as well as downward transport of H and O from above 85 km and a corresponding decrease in O3. This interpretation seems quite plausible. Overall, this is a very nice piece of work which illustrates the importance of using multi-instrumented observations. \\

We thank the reviewer for the assessment of our work and the positive remark.

\\ The first is there must be a statement somewhere acknowledging the limitations of making observations in a Eulerian framework. That is, you are not observing the same air mass over 8 hours. This means that your interpretation of events requires that the atmosphere is horizontally homogeneous over roughly 2000 km. Whereas, in fact you only know the degree of homeneity over about 35 km (the distance between the off-zenith lidar beams), with some additional information over a larger scale from the all-sky imager. There is nothing you can do about this, but it should be stated in the paper. \\

We agree with this point and as mentioned by the reviewer this is an unavoidable issue with ground based measurements. At the end of section 2, we have added the following

paragraph (revised manuscript will be uploaded after receiving all the comments during open discussion).

'Being ground based measurements made in the Eulerian framework, we cannot observe the same air mass for an extended period. Though we can observe the small-scale structures and their movements in the airglow images, they are also superposed with the background wind, which is derived from the lidar measurements in this work. While this is an unavoidable drawback in studying the atmosphere using ground based measurements, we assume that the processes occurring are sufficiently homogeneous in the horizontal directions.'

\\ The second issue is about the downward transport of H and O to below 85 km. From the way you describe this, the reader will imagine that the $NaHCO_3$ reservoir is left unchanged below 85 km, to be joined by O and H from aloft. However, the $NaHCO_3$ below 85 km will also be transported downwards. So it is actually a parcel of air containing $NaHCO_3$, H and O from above 85 km that is transported downwards and heats adiabatically, releasing Na. Note that the mixing ratio of total Na increases with height up to the ablation peak of Na which is above 90 km (see recent papers e.g. Carrillo-Sanchez et al., (2020), Icarus, 335, art. no. 113395 ). So downward transport will also increase the total Na concentration (i.e. Na + reservoir species) below 85 km. \\

Thanks for rising this issue. We have included this in many parts of the discussion section. Now we mention 'downward flux of minor species' in many places and retain downflux of H and O only where they are particularly discussed. The downward flux of $NaHCO_3$, Na and $O_3$ are also mentioned in the discussion part. The recent reference Carrillo-Sanchez et al., (2020) is also included.

\\ A third issue is that you list a large number of temperature-dependent rate coefficients, but do not do anything quantitative with them. That looks a little odd. For example, at line 390 you provide the rate coefficient for H + O3, and state that this

increases with temperature. But why not say how much? For example: "the rate coefficient increases by 40% when T increases from 200 to 230 K". That gives the reader some quantitative understanding of the point you are making. \\

In the revised version, we include Table 3 (attached at the end of this reply as a figure), which contains the values of the rate constants from 200 to 230 K in steps of 10 K and indicate the percentage increase in the rate constants. We believe that this will give a better understanding on the increased release of sodium atoms and a reduction in their reconversion to reservoir species. We also refer to the extent of variations in some parts of the Discussion section.

\\ One other point - although the paper is well written and straightforward to read, there are many grammatical errors - particularly the absence of the definite article "the" and indefinite article "a". It is not the job of a reviewer to correct these basic errors. \\

We are sorry for the grammatical errors. In the revised version, we have tried our level best to correct them and we are certain that most of the mistakes are corrected, if not all.

Minor corrections:

\\ line 23: change to: "...as a consequence of meteoric ablation (e.g. Plane ....)" \\

Changed.

\\ line 24: the statement "In high latitude winters, the peak altitudes are close to 88 km due to atmospheric circulation." is not really correct - it is chemistry which determines the height of the Na layer; the role of circulation is principally in changing the local temperature profile. \\

While it is true that the chemistry determines the equilibrium height of the Na layer not only in the high latitudes but in all the latitudes, the particular subsidence of peak height in winter polar region is believed to be due to the circulation and is observed with satellite based measurements (Fussen et al., ACP, 2010, in particular Figure 12).

[Figure]

\\ line 41: "occur at lower altitudes" \\

Corrected.

\\ line 73: "sodium lidar and airglow imaging observations from a high latitude location" \\

Corrected.

\\ line 79: in what way is the lidar "state of the art"? Please specify. The performance parameters you mention sound fairly standard. \\

The lidar is operated maintanence-free and uses a solid state laser diode end pumped Nd:Yag laser system to achieve high stability. The lidar functions without any manual adjustments required at the laser or telescope systems for the whole season as explained in Kawahara et al., Opt. Express, 2017. However, the lidar is being operated for the past 10 years and hence we remove the term 'state of the art' in the revised version. Instead, we include the above mentioned sentences to highlight the speciality of the lidar hardware.

\\ line 200: "This is further confirmed by the ..." \\

Changed.

\\ line 225: "...UT. The front continued ..." \\

Modified as suggested.

\\ line 269: "above 93 km before..." \\

Modified as suggested.

\\ line 314: "thermal ducting was possible" \\

Changed.

\\ line 319: "m and k stand for" \\

Corrected.

\\ line 384: "would have been initiated" \\

Changed.

\\ line 399: in fact, the Na compounds (NaOH, NaHCO3 etc.) photolyse in the near-UV above 200 nm. So change EUV to UV. \\

Changed. Thank you for this information.

\\ line 446: "have led to " \\

Changed.

\\ The References need to be sorted out and checked. They are not all in alphabetical order, and the same author appears with different initials in difference references! \\

We apologize for this mistake. We have extracted the references in bibtex format from the journal websites and created the list. We have checked and corrected the mistakes in the revised version (will be uploaded after the discussion closes).

We once again thank the reviewer for the evaluation of the work and useful suggestions that led to its improvement.

———————————————

| Reaction | Rates from 200 to 230 K | | | | % increase with respect to 200 K | | | |
|---|---|---|---|---|---|---|---|---|
| | 200 K | 210 K | 220 K | 230 K | 200 K | 210 K | 220 K | 230 K |
| R1 | $1.34 \times 10^{-11}$ | $1.49 \times 10^{-11}$ | $1.65 \times 10^{-11}$ | $1.81 \times 10^{-11}$ | 0 | 12 | 24 | 36 |
| R2 | $7.09 \times 10^{-14}$ | $9.38 \times 10^{-14}$ | $1.21 \times 10^{-13}$ | $1.53 \times 10^{-13}$ | 0 | 32 | 71 | 116 |
| R3 | $2.56 \times 10^{-12}$ | $2.91 \times 10^{-12}$ | $3.28 \times 10^{-12}$ | $3.66 \times 10^{-12}$ | 0 | 14 | 28 | 43 |
| R4 | $2.20 \times 10^{-10}$ | $2.25 \times 10^{-10}$ | $2.31 \times 10^{-10}$ | $2.36 \times 10^{-10}$ | 0 | 2 | 5 | 7 |
| R5 | $6.16 \times 10^{-10}$ | $6.33 \times 10^{-10}$ | $6.49 \times 10^{-10}$ | $6.64 \times 10^{-10}$ | 0 | 3 | 5 | 8 |
| R6 | $5.00 \times 10^{-30}$ | $4.71 \times 10^{-30}$ | $4.45 \times 10^{-30}$ | $4.22 \times 10^{-30}$ | 0 | -6 | -11 | -16 |
| R7 | $1.59 \times 10^{-33}$ | $1.41 \times 10^{-33}$ | $1.26 \times 10^{-33}$ | $1.14 \times 10^{-33}$ | 0 | -11 | -20 | -28 |
| R8 | $2.69 \times 10^{-16}$ | $4.39 \times 10^{-16}$ | $6.86 \times 10^{-16}$ | $1.03 \times 10^{-15}$ | 0 | 63 | 155 | 283 |

**Fig. 1.** Table 3

---

## Author Comment (AC2) · 7 Dec 2020

Response to the reviewer 2 comments on the manuscript 'Formation of a bottomside secondary sodium layer associated with the passage of multiple mesospheric frontal systems'

We thank the reviewer for going through our work and providing suggestions on the same. Here we address the concerns raised by the reviewer and indicate the necessary modifications made in the revised version of the manuscript. Reviewer's com-

ments are given between double backslashes and our responses are below the comments. (i.e \\ Reviewer Comment \\ Our replies ).

\\ This paper studies the formation mechanism of the secondary Na peak that appears within the altitude range of the main Na layer but below the main peak and near the bottomside of the main layer. The data quality (both lidar and airglow imager near Tromso) is high, and the analyses connecting the lidar-observed secondary Na peak below its main layer peak with the OH-imager-observed mesospheric bore event are extensive. The topic is interesting to the middle atmosphere science community. On this aspect, the paper is worth considering for publication in ACP after extensive reviews and revisions.

However, there are three major issues with the current manuscript: 1) The paper title is misleading or improper, 2) its Abstract reads badly with the first sentence distract people's attention, and 3) its Introduction contains misunderstanding of metal layer sciences.

All these issues likely stem from authors' misunderstanding of the meteoric metal layers. The main Na layer ranges from _75 to 110 km, and the layer (below 85 km) they reported here is well within the main Na layer. Therefore, it is NOT an extra layer to the main layer, but an extra peak to the main layer peak. We have seen many times that Na layers go well below 85 km forming variable peaks during wintertime in the polar region, most likely caused by various wave activity. Therefore, what authors observed isn't new, but their studies of connecting such Na peaks to bore/frontal events are new and worth publishing. \\

We thank the reviewer for the positive opening remark. Though the peaks of sodium layer occasionally form below 85 km, in this case we find a clear separation between the main peak and the lower peak and hence referred it as layer. Now the terminology is changed to 'peak'. The reviewer may also note that we never claimed that the observation of lower altitude sodium layer is new. Below we have addressed the three

issues pointed by the reviewer.

\\ 1) Paper title: First, as written above it's not a new Na layer, but it's the secondary Na peak within the main Na layer; second, "in lower altitudes" has a grammar issue – lower than what? Therefore, such a paper title is not acceptable. Authors may consider to change the paper title to "Formation of an extra Na peak below the main layer peak associated with passage of multiple mesospheric frontal system" or something better.\\

The reviewer may note that, while the sodium layer exists between the altitude region of 75 and 110 km, it does not fill the whole range of altitudes. Often, more than one peak is observed within the altitude range. When the different peaks are well separated, they can be referred as 'layers' in our opinion. The reviewer may kindly note that the term 'layer' is used for clearly distinguishable peaks in the sodium concentration in the context of sporadic sodium layers, most of which occur within the altitude range of 75 to 110 km. Nevertheless, respecting the opinion of the reviewer and following the suggestion to change the title, we have changed the title of the manuscript as 'Formation of an additional density peak in the bottomside sodium layer associated with the passage of multiple mesospheric frontal systems'.

Regarding 'in lower altitudes': There is no such term in the title. In the updated version of the manuscript, we clearly mention in the text that: 'We refer to the peak occurring below the main sodium layer peak at 90 km as 'lower altitude sodium peak' in this work'. The updated manuscript version will be uploaded in a few days.

\\ 2) Abstract: The first sentence in the Abstract is very misleading and it is frustrating to read it. Your paper is on the secondary Na peak below the main layer peak, but you started with mentioning something that is non-relevant to your subject. Please remove this sentence to avoid misleading readers. Also, change "additional sodium layer" to "additional Na peak". \\

The first sentence of the abstract is changed following the reviewer's suggestion as

'We present a detailed investigation of the formation of an additional sodium density peak at altitudes of 79-85 km below the main peak of sodium layer based on sodium lidar and airglow imager measurements made at Ramfjordmoen near Tromsø, Norway on the night of 19 December 2014.'

While we believe that it is not a mistake to call such a separated peak as an additional layer, we understand the concern of the reviewer that on many other occasions, there are peaks below 85 km that are not separated from the main layer to the extent of the present case. Therefore, we change the terminology from 'lower altitude sodium layer' to 'lower altitude sodium peak' as suggested. As mentioned in the response of previous comment, we also explain in the manuscript text what is 'lower altitude' in the context of this work.

\\ 3) Introduction: There is some lack of understanding of the thermosphere-ionosphere metal (TIMt) layers (mentioned in review paper by Plane et al. (2015)) in the Introduction, including thermosphere-ionosphere Fe and Na (TIFe and TINa) layers that were discovered to reach the altitudes of lower F region (Chu et al., GRL, 2011, 2020; Tsuda et al., GRL, 2015; Raizada et al., 2015; Chu and Yu, 2017). None of these pioneering papers were acknowledged. Instead, authors referenced Collins et al. (1996) and Wang et al. (2012), and adapted a bad phrase "double sodium layers". This "double sodium layers" phrase is improper and misleading, thereby it has been discarded by the field. Therefore, the current paragraph (the 3rd one in Introduction) is totally not acceptable. However, these TIMt layers aren't the focus of this manuscript, so authors may choose to remove this paragraph entirely and focus on the main Na layer. If authors want to include TIMt layers in the introduction, then they should update their understanding of the TIMt layers and cite proper references \\

We are sorry but we do not find this comment relevant to the present manuscript. We believe this comment was already addressed during the access peer review stage of the manuscript. The third paragraph of previous version was already removed before the manuscript was put in to discussion. The third paragraph of the current manuscript

explains the research problem addressed in the work. We never use the term 'double sodium layer' in the whole manuscript. Moreover, we note that all the references except Chu et al., 2020 and Chu and Yu, 2017 are already present in the manuscript. We now include Chu et al., 2020 which is recent and we were unaware of earlier. The reference Chu and Yu, 2017 discusses only about Fe/Fe+ in the thermosphere while our manuscript discusses about Na in the mesosphere. Therefore, we are unable to include the reference and sorry about that.

We once again thank the reviewer for assessing the work and providing suggestions for the improvement of the manuscript. We hope we have satisfactorily addressed the concerns.

---

## Referee Report (RR1)

**3rd round review on "Formation of an additional density peak in the bottomside of sodium layer associated with the passage of multiple mesospheric frontal systems" by Narayanan et al. to ACP**

After addressing two rounds of review comments, authors have improved the quality of this paper. The lidar-observed additional Na peak below the main layer peak and the OH-imager-observed mesospheric frontal systems are interesting phenomena. Authors explained the observations using downward transport of Na species, H, and O by the mesospheric bores as well as with the enhanced temperatures, which is quite reasonable. The data and explanation may inspire future modeling and observations. However, there are still some issues with the paper on three main aspects: 1) Authors still have some misunderstanding of the thermosphere-ionosphere metal layers versus the sporadic Na layers. 2) The vertical wind data have significant bias, which is not reasonable. 3) Horizontal advection should be mentioned as another possibility to explain part of the observations. There are also numerous grammar issues.

Therefore, I would like to recommend the paper for publication in ACP after authors address the following comments that go by page numbers:

1) Page 1, line 15: Change "This would have liberated …" to "**Both factors** would have liberated …"

2) Page 2, line 30: Change "They form due to the wind shears collecting …" to "They form due to the **wind shear mechanisms** collecting …". Note that wind shears themselves cannot accumulate ions or atoms, but it is the wind shear mechanisms via $\vec{V} \times \vec{B}$ to collect ions.

3) Page 2, lines 32-39: **Authors should move Collins et al. (1996) reference from line 38 to line 32**; that is, it is a reference for sporadic Na layers (SSLs) but not a reference for the thermosphere-ionosphere metal (TIMt) layers. It is necessary to recognize that Collins et al. (1996) paper reported high-altitude sporadic Na layers, which were NOT the TIMt layers; therefore, this paper should be moved to line 32 along with Cox and Plane (1998) etc., and removed from line 38 (Chu et al., 2011; etc.). For authors information, the high-altitude SSLs reported in Collins et al. (1996) are very similar to the sporadic Fe layers around 110 km from 18 UT to 21 UT in Figure 1 of Chu et al. (2011), but they are very different from the thermospheric metal layers reported by Chu et al. (2011), Wang et al. (2012), etc. Authors should pay more attention in referencing proper papers at proper places.

4) Section 2 "Data used" – this section is very long while still not informative enough. For example, it is unclear what resolutions were used in the temperature data retrieval. Can authors use a table to tabulate related information but shorten the section text? Descriptions on how lidar data were retrieved and how OH images were analyzed are quite lengthy but aren't they standard procedures? Anything new authors developed? If not new things, why don't you reference some papers and then shorten the description?

If authors feel strong to tell readers how they handled the data for certain purpose, why don't you put such contents to Section 3 when related results are presented. Otherwise, it is a bit frustrating to read the lengthy Section 2 before knowing what results you got.

5) Page 9, line 210: change "creation of sodium atoms" to "**production** of sodium atoms"

6) Page 9-10, Figure 3: Authors wrote "This is further confirmed by the observation that the column abundance was reduced after the disappearance of the lower altitude sodium peak". However, the Na abundance level near the end of the observation was higher than that at the beginning of the observation, i.e., the Na layer did not return to the original state after the passing of frontal systems. Therefore, **it is necessary to show how Na column abundance changes through a normal night without mesospheric frontal systems**, which will check whether the increase of column abundance during the frontal systems is unusual when compared to a normal night.

7) Page 10, line 230: Change "horizon" to "edge of the image"

8) Page 12, Figure 5: What is the reference point for "Distance" in the x-axis label, i.e., "Distance" from which point?

9) Page 13, line 263: Change "Now we discuss" to "Now we **present**".
Page 13, line 264: Change "The temperature profile" to "The temperature **contour**"

10) Page 14, Figure 7: The color scales for $N^2$ plot are unclear – does the blue color represent negative $N^2$ or not?

11) Page 15, Figure 10: **The vertical wind data is unacceptable** because it shows a very large negative wind bias. Majority of the vertical winds are between 0 m/s and -10 m/s, which cannot be true for the real atmosphere. It appears that the Na lidar on 19 Dec 2014 exhibited a large pulsed laser frequency offset (or frequency chirp), and authors didn't correct the frequency offset – leading to the negative bias in the results. Authors should either correct the vertical wind data or remove the vertical wind plots from the paper – the current Figure 10 top plot is unacceptable.

Also, the higher values of vertical velocities near 85 km (line 293) appear to be dominated by noise or measurement errors. Authors should be really careful in using the vertical wind data or in the interpretation.

12) Page 21, Figure 14: For the integrated densities from 88-95 km, they are positively correlated with the mean temperature quite well up to 17 UT, but then the correlation becomes negative. This result makes me wondering whether some of the variations are caused by the horizontal advection of the Na layers. This factor should be mentioned on page 25 in the paragraph above line 480.

13) Page 22, line 411-413: How can Na layer peak affect the formation of bores?

14) Page 24, line 444: Change "The principle loss" to "The **principal** loss"

15) Page 25, line 472: Change "…the bores have lead to …" to "…the bores have **led** to …"

**1) Scientific significance -- Excellent**
Does the manuscript represent a substantial contribution to scientific progress within the scope of this journal (substantial new concepts, ideas, methods, or data)?

**2) Scientific quality – Good**
Are the scientific approach and applied methods valid? Are the results discussed in an appropriate and balanced way (consideration of related work, including appropriate references)?

**3) Presentation quality -- Good**
Are the scientific results and conclusions presented in a clear, concise, and well structured way (number and quality of figures/tables, appropriate use of English language)?

---

## Author Response (AR2)

**Response to Reviewer's comments on our revised manuscript:**

We thank the reviewer for the useful comments. Below, we give our response to the reviewer's comments. Comments are given between double backslashes followed by our responses.
(i.e.   \\ reviewer comments\\
Our response )

\\ After addressing two rounds of review comments, authors have improved the quality of this paper. The lidar-observed additional Na peak below the main layer peak and the OH-imager-observed mesospheric frontal systems are interesting phenomena. Authors explained the observations using downward transport of Na species, H, and O by the mesospheric bores as well as with the enhanced temperatures, which is quite reasonable. The data and explanation may inspire future modeling and observations. However, there are still some issues with the paper on three main aspects: 1) Authors still have some misunderstanding of the thermosphere-ionosphere metal layers versus the sporadic Na layers. 2) The vertical wind data have significant bias, which is not reasonable. 3) Horizontal advection should be mentioned as another possibility to explain part of the observations. There are also numerous grammar issues. Therefore, I would like to recommend the paper for publication in ACP after authors address the following comments that go by page numbers:\\

We thank the reviewer for the review comments and positive opening remark. We address the comments below and made necessary modifications to the manuscript based on the review comments.

\\1) Page 1, line 15: Change "This would have liberated …" to "**Both factors** would have liberated …"\\

Modified as suggested

\\ 2) Page 2, line 30: Change "They form due to the wind shears collecting …" to "They form due to the **wind shear mechanisms** collecting …". Note that wind shears themselves cannot accumulate ions or atoms, but it is the wind shear mechanisms via $V \times B$ to collect ions.\\

As per suggestion changed to 'They form due to the wind shear mechanism producing ion convergence in a narrow altitude range'.

\\ 3) Page 2, lines 32-39: **Authors should move Collins et al. (1996) reference from line 38 to line 32**; that is, it is a reference for sporadic Na layers (SSLs) but not a reference for the thermosphere-ionosphere metal (TIMt) layers. It is necessary to recognize that Collins et al. (1996) paper reported high-altitude sporadic Na layers, which were NOT the TIMt layers; therefore, this paper should be moved to line 32 along with Cox and Plane (1998) etc., and removed from line 38 (Chu et al., 2011; etc.). For authors information, the high-altitude SSLs reported in Collins et al. (1996) are very similar to the sporadic Fe layers around 110 km from 18 UT to 21 UT in Figure 1 of Chu et al. (2011), but they are very different from the thermospheric metal layers reported by Chu et al. (2011), Wang et al. (2012), etc. Authors should pay more attention in referencing proper papers at proper places.\\

The reference is moved to line 34 along with Cox and Plane, 1998 etc.

\\ 4) Section 2 "Data used" – this section is very long while still not informative enough. For example, it is unclear what resolutions were used in the temperature data retrieval. Can authors use a table to tabulate related information but shorten the section text? Descriptions on how lidar data were retrieved and how OH images were analyzed are quite lengthy but

aren't they standard procedures? Anything new authors developed? If not new things, why don't you reference some papers and then shorten the description?
If authors feel strong to tell readers how they handled the data for certain purpose, why don't you put such contents to Section 3 when related results are presented. Otherwise, it is a bit frustrating to read the lengthy Section 2 before knowing what results you got. \\

The reviewer may note that we did not discuss the lidar data retrieval in this section. Previous works are referred for the data retrieval methods (Nozawa et al., 2014 and Kawahara et al., 2017). We only mentioned the essential information like spatial and temporal resolutions of the raw data. Rest of the section describes the type of averaging we made, and elaborates on calculation of other relevant parameters like Buoyancy frequency. If any interested researcher intends to reproduce the results in future, they need to know how each step is carried out and hence we prefer to keep the current section.

However, following the comment no. 11 of the reviewer, we have removed Figure 10 of the previous version showing vertical winds and Richardson numbers. This helped in removing the part of text describing the vertical winds and calculation of Richardson numbers in section 2. Length of Section 2 is reduced due to this modification.

\\ 5) Page 9, line 210: change "creation of sodium atoms" to "**production** of sodium atoms" \\

Changed

\\ 6) Page 9-10, Figure 3: Authors wrote "This is further confirmed by the observation that the column abundance was reduced after the disappearance of the lower altitude sodium peak". However, the Na abundance level near the end of the observation was higher than that at the beginning of the observation, i.e., the Na layer did not return to the original state after the passing of frontal systems. Therefore, **it is necessary to show how Na column abundance changes through a normal night without mesospheric frontal systems**, which will check whether the increase of column abundance during the frontal systems is unusual when compared to a normal night. \\

The reviewer may please refer to Figure 12 (Figure 13 in the previous version) and the discussion in the lines 355 – 362. We quantify that about 65% of the column abundance increase is due to the formation of lower level peak and the remaining enhancement is due to the increased sodium concentration in the main sodium layer and its topside. We mentioned 'column abundance was reduced' since we never expect that to be the same value prior to the passage of the bores. This is because there are other processes that can bring about the sodium abundance variations. As the reviewer may well be aware of, the column abundance often varies with tidal and gravity wave variations. Since we never claimed that we identified the mesospheric fronts with the column abundance variations, we feel it is not necessary to include column abundance figure from some other normal night. Another problem is how to define 'normal night'? One day may have significantly stronger tide and another day may have an intense gravity wave and so on. Also, our understanding on how the mesospheric sodium concentrations respond to different level of auroral activity is not complete. Tromsø being auroral site, defining 'normal night' is not an easy task.

However, we attach a Figure showing the column abundances for 18, 19 and 21 December 2014 to our response (19 December being day of this study). It may be seen that other nearby days also show fluctuations in the column abundance but they did not reveal bores to our knowledge. In the present case the mentioned enhancement in column abundance coincided with formation of the lower peak which is caused by successive passage of bores. We don't include this Figure to the manuscript as we feel this is not required for the present study.

[Figure]

\\ 7) Page 10, line 230: Change "horizon" to "edge of the image" \\

Changed

\\ 8) Page 12, Figure 5: What is the reference point for "Distance" in the x-axis label, i.e., "Distance" from which point? \\

Please see lines 174 to 180 describing about the distance axis for extracted cross sections. The distance is from the starting point of the extracted cross section.

\\ 9) Page 13, line 263: Change "Now we discuss" to "Now we **present**". Page 13, line 264: Change "The temperature profile" to "The temperature **contour**" \\

Both the changes are made.

\\ 10) Page 14, Figure 7: The color scales for N^2 plot are unclear – does the blue color represent negative N^2 or not? \\

The color scales are chosen after trial and error method to clearly reveal the duct location. More contours may bring confusion to the readers. The dark blue represent negative N^2. Blue does not represent negative values. Please refer to the colorscale added to the right.

\\ 11) Page 15, Figure 10: **The vertical wind data is unacceptable** because it shows a very large negative wind bias. Majority of the vertical winds are between 0 m/s and -10 m/s, which cannot be true for the real atmosphere. It appears that the Na lidar on 19 Dec 2014 exhibited

a large pulsed laser frequency offset (or frequency chirp), and authors didn't correct the frequency offset – leading to the negative bias in the results. Authors should either correct the vertical wind data or remove the vertical wind plots from the paper – the current Figure 10 top plot is unacceptable.
Also, the higher values of vertical velocities near 85 km (line 293) appear to be dominated by noise or measurement errors. Authors should be really careful in using the vertical wind data or in the interpretation. \\

We sincerely thank the reviewer for noticing this bias and pointing it out. We agree that there is a bias. We have removed Figure 10 from the revised version. The vertical winds and Richardson number estimations are not critical for explaining our results. We referred to the Richardson number only once in the discussion. Hence we removed Figure 10 following the suggestion of the reviewer and it also helps in reducing the length of the manuscript, in particular length of section 2 (See comment no. 4).

\\ 12) Page 21, Figure 14: For the integrated densities from 88-95 km, they are positively correlated with the mean temperature quite well up to 17 UT, but then the correlation becomes negative. This result makes me wondering whether some of the variations are caused by the horizontal advection of the Na layers. This factor should be mentioned on page 25 in the paragraph above line 480.¨ \\

Based on the reviewer's comment, we have added the following statement in lines 385 – 389 where we discuss Figure 14 (Figure 13 in the present version).
'On the other hand, in the presented observations after 17:15 UT and between 88 and 95 km, the sodium density variations do however not correlate with temperature. This may be either due to the horizontal advection of sodium atoms or due to the ion chemistry as this time also coincides with onset of aurora.'

\\ 13) Page 22, line 411-413: How can Na layer peak affect the formation of bores? \\

Please note that we never mentioned that Na layer peak affect the formation of bores. We meant that the enhanced strong thermal ducting occurred coincident with the region of the main sodium layer. The sentence is slightly modified to avoid confusion while reading.

\\ 14) Page 24, line 444: Change "The principle loss" to "The **principal** loss" \\

Changed.

\\ 15) Page 25, line 472: Change "…the bores have lead to …" to "…the bores have **led** to …" \\
Corrected.

We believe that we have satisfactorily addressed the comments of the reviewer. We once again thank the reviewer for the review, in particular pointing to the bias in the vertical wind.
We also thank the Editor for handling the manuscript.